# Contrary neuronal recalibration in different multisensory cortical areas

Fu Zeng[1], Adam Zaidel[2]*, Aihua Chen[1]*

[1]Key Laboratory of Brain Functional Genomics (Ministry of Education), East China Normal University, Shanghai, China; [2]Gonda Multidisciplinary Brain Research Center, Bar-Ilan University, Ramat Gan, Israel

**Abstract** The adult brain demonstrates remarkable multisensory plasticity by dynamically recalibrating itself based on information from multiple sensory sources. After a systematic visual–vestibular heading offset is experienced, the unisensory perceptual estimates for subsequently presented stimuli are shifted toward each other (in opposite directions) to reduce the conflict. The neural substrate of this recalibration is unknown. Here, we recorded single-neuron activity from the dorsal medial superior temporal (MSTd), parietoinsular vestibular cortex (PIVC), and ventral intraparietal (VIP) areas in three male rhesus macaques during this visual–vestibular recalibration. Both visual and vestibular neuronal tuning curves in MSTd shifted – each according to their respective cues' perceptual shifts. Tuning of vestibular neurons in PIVC also shifted in the same direction as vestibular perceptual shifts (cells were not robustly tuned to the visual stimuli). By contrast, VIP neurons demonstrated a unique phenomenon: both vestibular and visual tuning shifted in accordance with vestibular perceptual shifts. Such that, visual tuning shifted, surprisingly, contrary to visual perceptual shifts. Therefore, while unsupervised recalibration (to reduce cue conflict) occurs in early multisensory cortices, higher-level VIP reflects only a global shift, in vestibular space.

## Editor's evaluation

This important study combines quantitative behavior and single-unit recordings in nonhuman primates to investigate the role of three cortical areas in cross-modal sensory calibration, a form of neural plasticity that is important for perception and learning. The results convincingly demonstrate key similarities and striking differences across the three areas and provide the first evidence for this form of calibration (in correspondence with behavior) at the level of single neurons. The work will be of broad interest to neuroscientists and psychologists studying multisensory perception, plasticity, and the role of sensory and association cortices in perceptual decisions.

*For correspondence:
adam.zaidel@biu.ac.il (AZ);
ahchen@brain.ecnu.edu.cn (AC)

**Competing interest:** The authors declare that no competing interests exist.

## Introduction

Our different sensory systems each continuously adapt to changes in the environment (*Webster, 2012*). Thus, to maintain stable and coherent perception in a multisensory and ever-changing world, the brain needs to dynamically adjust for sensory discrepancies between the different modalities. This process of multisensory recalibration takes place continually and is perhaps more fundamental than multisensory integration because integration would not be beneficial when the underlying cues are biased. While the neural bases of multisensory integration have received a lot of attention (*Chen et al., 2013a*; *Gu et al., 2008*; *Stein et al., 2014*), the neural bases of multisensory recalibration have been explored to a much lesser degree.

Cross-modal recalibration has been observed in a variety of multisensory settings. One well-known example is the ventriloquist aftereffect, in which exposure to a consistent spatial discrepancy between

auditory and visual stimuli induces a subsequent shift in the perceived location of sounds (*Bertelson and De Gelder, 2004*; *Canon, 1970*; *Kramer et al., 2019*; *Radeau and Bertelson, 1974*; *Recanzone, 1998*; *Watson et al., 2021*). Also, the rubber-hand illusion leads to an offset in hand proprioception in the direction of the visually observed rubber hand (*Abdulkarim et al., 2021*; *Botvinick and Cohen, 1998*; *Kennett et al., 2001*; *Thériault et al., 2022*; *Tsakiris and Haggard, 2005*). Although it was initially thought that only the non-visual cues recalibrate to vision, termed visual dominance (*Brainard and Knudsen, 1993*; *Rock and Victor, 1964*), further work in a variety of paradigms has revealed both visual and non-visual recalibration (*Atkins et al., 2003*; *Lewald, 2002*; *van Beers et al., 2002*; *Zaidel et al., 2011*).

Most of what we know about multisensory recalibration is described at the behavioral level (*Lewald, 2002*), with little known about its neuronal underpinnings. Recent studies in humans have shed some light on this question. In the ventriloquism aftereffect, cross-modal (audio-visual) recalibration of auditory signals (fMRI) is seen in low-level auditory cortical areas (*Zierul et al., 2017*). According to that study and another recent (EEG) study (*Park and Kayser, 2021*), higher-level parietal regions also play a central role in cross-modal spatial recalibration. Moreover, *Park and Kayser, 2021* further suggest that frontal regions consolidate the behavioral shift under sustained multisensory discrepancies. However, these methods (fMRI and EEG) lack the resolution to probe recalibration at the level of single neurons.

In a series of classic studies, Kundsen and Brainard investigated multisensory plasticity at the neuronal and circuit levels in the barn owl (*Knudsen, 2002*; *Knudsen and Brainard, 1991*; *Linkenhoker and Knudsen, 2002*). They found profound neuronal plasticity in juvenile owls reared with prismatic lenses that systematically displaced their field of view. In that case, the auditory space map in the optic tectum was recalibrated to be aligned with the displaced visual field (*Knudsen and Brainard, 1991*). However, multisensory plasticity is not limited to the development, and the neuronal bases of how multiple sensory systems continuously adapt to one another in the adult brain remain fundamentally unknown.

Self-motion perception (the subjective feeling of moving through space) relies primarily on visual and vestibular cues (*Butler et al., 2015*; *Butler et al., 2010*; *de Winkel et al., 2010*; *Fetsch et al., 2011*; *Fetsch et al., 2009*; *Gu et al., 2007*; *Warren et al., 1988*). Multisensory integration of visual and vestibular signals can improve heading perception (*Butler et al., 2015*; *Dokka et al., 2015*; *Gu et al., 2008*). However, conflicting or inconsistent visual and vestibular information often leads to motion sickness (*Oman, 1990*; *Reason and Brand, 1975*). Interestingly, this subsides after prolonged exposure to the sensory motion conflict, presumably through brain mechanisms of multisensory recalibration (*Held, 1961*; *Shupak and Gordon, 2006*). Thus, self-motion perception – a vital skill for everyday function with intrinsic plasticity – offers a prime substrate to study crossmodal recalibration.

We previously investigated and found robust perceptual recalibration of both visual and vestibular cues in response to a systematic vestibular–visual heading discrepancy (*Zaidel et al., 2011*). Similar results were seen for both humans and monkeys. In that paradigm, no external feedback was given. Thus, the need for recalibration arose solely because of the cue discrepancy (we therefore call this condition *unsupervised*). This led to shifts in subsequent visual and vestibular perceptual estimates toward each other, presumably to reduce the conflict. This is in line with the notion that unsupervised recalibration aims to maintain 'internal consistency' between the cues (*Burge et al., 2010*). However, the neuronal basis of this everyday multisensory plasticity is unknown. This study aimed to test unsupervised recalibration of visual and vestibular neuronal tuning, and how it may differ across multisensory cortical areas.

In line with human neuroimaging studies that showed cross-modal (auditory–visual) recalibration in relatively early sensory areas (*Amedi et al., 2002*; *Zierul et al., 2017*), and because unsupervised recalibration is sensory driven (occurs as a result of the cross-modal discrepancy, in the absence of overt feedback) we expected to observe neural correlates of unsupervised visual–vestibular recalibration in relatively early cortical areas with self-motion signals. Previous studies with monkeys identified two relatively early multisensory cortical areas involved in self-motion perception: the medial superior temporal area (*Gu et al., 2006*) and the parietal insular vestibular cortex (PIVC, *Chen et al., 2010*). Neurons in MSTd respond to large optic flow stimuli, conducive to the visual perception of self-motion (*Gu et al., 2006*). Vestibular responses are also present in MSTd, however visual self-motion signals

dominate (*Gu, 2018*; *Gu et al., 2012*). PIVC has strong vestibular responses, without strong tuning to visual optic flow (*Chen et al., 2010*).

The ventral intraparietal (VIP) area also has robust responses to visual and vestibular self-motion stimuli, however, it is marked by strong choice signals (*Chen et al., 2016*; *Gu, 2018*; *Zaidel et al., 2017*). It is thus considered a higher-level multisensory area, possibly involved in perceptual decision-making or higher-order perceptual functions. Accordingly, and in line with findings of parietal involvement in human cross-modal recalibration (*Park and Kayser, 2021*; *Zaidel et al., 2021*; *Zierul et al., 2017*), we were interested to see what correlates of unsupervised recalibration we would see in parietal neurons. Different types of multisensory recalibration observed in VIP vs. lower-level (MSTd and PIVC) multisensory areas can provide important insights into their differential underlying functions. Thus, in this study, we focused on these three multisensory cortical areas. We examined whether and how their visual and vestibular neural tuning changed in accordance with corresponding perceptual shifts during a single session (~1 hr) of unsupervised cross-modal recalibration.

## Results

Three monkeys performed a heading discrimination task before, during, and after undergoing cross-modal recalibration to spatially conflicting vestibular–visual signals. Simultaneous to behavioral performance, we recorded from single neurons extracellularly in areas MSTd (upper bank of the superior temporal sulcus, $N$ = 83 total; 19 from monkey D, 64 from monkey K), PIVC (upper bank and the tip of the lateral sulcus, $N$ = 160 total; 91 from monkey D, 69 from monkey B), and VIP (lower bank and tip of the intraparietal sulcus, $N$ = 118 total; 103 from monkey D, 15 from monkey B).

The experiment paradigm followed similar methodology as our previous behavioral study (*Zaidel et al., 2011*). It consisted of three consecutive blocks: pre-recalibration (*Figure 1A*), recalibration (*Figure 1B*), and post-recalibration (*Figure 1C*). In the recalibration block, the monkeys were presented with combined stimuli (simultaneous visual and vestibular cues) with a systematic discrepancy between the visual and vestibular heading directions. In the pre- and post-recalibration blocks, the unisensory perception was measured using visual-only or vestibular-only cues. The effects of recalibration on visual and vestibular perception were measured by the shifts in the post- vs. pre-recalibration psychometric curves. We first (in the next section) present the monkeys' perceptual recalibration results. Thereafter, we present the neural correlates thereof.

### Vestibular and visual perceptual estimates shift toward each other

*Figure 2* shows example psychophysical data from two experimental sessions. Replicating our previous behavioral results (*Zaidel et al., 2011*), we found that both visual and vestibular psychometric functions shifted in the direction required to reduce cue conflict. Namely, when the vestibular and visual heading stimuli were systematically offset, such that they consistently deviated to the right and the left, respectively ($\Delta^+$, *Figure 2A*), the vestibular post-recalibration curve (blue) was shifted rightward vs. pre-recalibration (black). Note that a rightward shift of the psychometric curve indicates a *leftward* perceptual shift (identified by a lower propensity for 'rightward' choices at 0° heading for the blue curve). Complementarily, the visual post-recalibration psychometric curve (red) shifted leftward vs. pre-recalibration (black), albeit to a lesser degree, indicating a *rightward* perceptual shift. In a reverse manner, when the vestibular and visual heading stimuli were offset to the left and right, respectively ($\Delta^-$, *Figure 2B*), the vestibular post-recalibration curve (blue) shifted to the left, and the visual post-recalibration curve shifted to the right.

These perceptual shifts were quantified by the difference between the post- vs. pre-recalibration curves' PSEs (points of subjective equality). A psychometric curve's PSE represents the heading angle of equal right/left choice proportion, that is, the heading that participants would supposedly perceive as straight ahead. The vestibular and visual psychometric shifts were positive and negative, 3.40° and −1.01°, respectively, in *Figure 2A*, and negative and positive, −3.68° and 1.00°, respectively, in *Figure 2B*. Thus, in both cases (*Figure 2A, B*), both the vestibular and the visual cues shifted in the direction required to reduce the cue conflict (i.e., in opposite directions).

In each session, only one discrepancy orientation was tested. Namely, vestibular and visual headings were either offset to the right and left (respectively), or vice versa. These discrepancies were arbitrarily defined as positive ($\Delta^+$) or negative ($\Delta^-$), respectively. In total, we collected data from 241

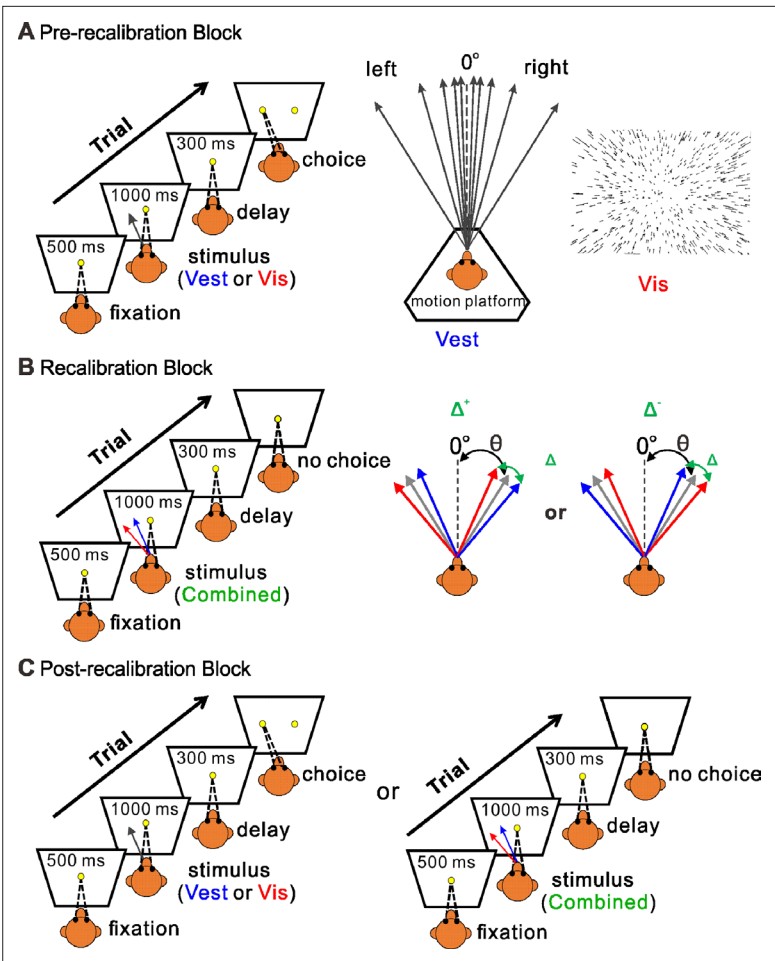

**Figure 1.** Multisensory recalibration paradigm. (**A**) Pre-recalibration block. The vestibular stimulus was elicited by moving the motion platform (schematic in the middle, viewed from above). The visual stimulus, presented on the screen in front of the monkey, corresponded to optic flow (schematic on the right) as it would be experienced during self-motion (without motion of the platform). The self-motion stimuli comprised linear motions (of either vestibular or visual stimuli) in a forward motion with a small leftward or rightward component (black arrows, schematic in the middle). Monkeys were required to fixate on a central target (yellow circle) presented on the screen during the stimulus and then to report their perceived heading by making a saccade to one of two targets (left or right relative to straight ahead). The heading angle ($\theta$) was varied across trials. (**B**) Recalibration block. Simultaneous vestibular and visual stimuli (combined) with a systematic discrepancy ($\Delta$) between the vestibular and visual headings were presented. Only one discrepancy orientation ($\Delta^+$ or $\Delta^-$) was used per session. The blue and red arrows represent the vestibular and visual headings, respectively. The gray arrows represent the headings (varied across trials) from which the vestibular and visual cues were offset (to either side by $\Delta/2$). The black dashed lines represent straight ahead. (**C**) Post-recalibration block. The single-cue trials (like in A) were interleaved with combined-cue trials (like in B).

sessions with $\Delta^+$ and 227 sessions with $\Delta^-$. Distributions of the vestibular and visual PSE shifts across sessions are presented in *Figure 2C* (above and below the abscissa for $\Delta^+$ and $\Delta^-$, respectively). The vestibular PSEs were shifted significantly to the right for the $\Delta^+$ condition (mean ± SE = 1.12° ± 0.12°; p = 2.3 × 10^{-17}, paired *t*-test), and significantly to the left for the $\Delta^-$ condition (mean ± SE = −1.76° ± 0.14°; p = 1.0 × 10^{-28}, paired *t*-test). The visual PSEs were shifted significantly to the left for the $\Delta^+$ condition (mean ± SE = −0.73° ± 0.11°; p = 6.5 × 10^{-11}, paired *t*-test), and significantly to the right for the $\Delta^-$ condition (mean ± SE = 1.10° ± 0.10°; p = 5.4 × 10^{-23}, paired *t*-test). Thus, consistent with our previous study, both cues shifted (in opposite directions) to reduce cue conflict.

Comparing the vestibular vs. visual shift magnitudes (pooled by flipping the vestibular and visual shift signs in the $\Delta^-$ and $\Delta^+$ conditions, respectively) demonstrated significantly larger vestibular vs.

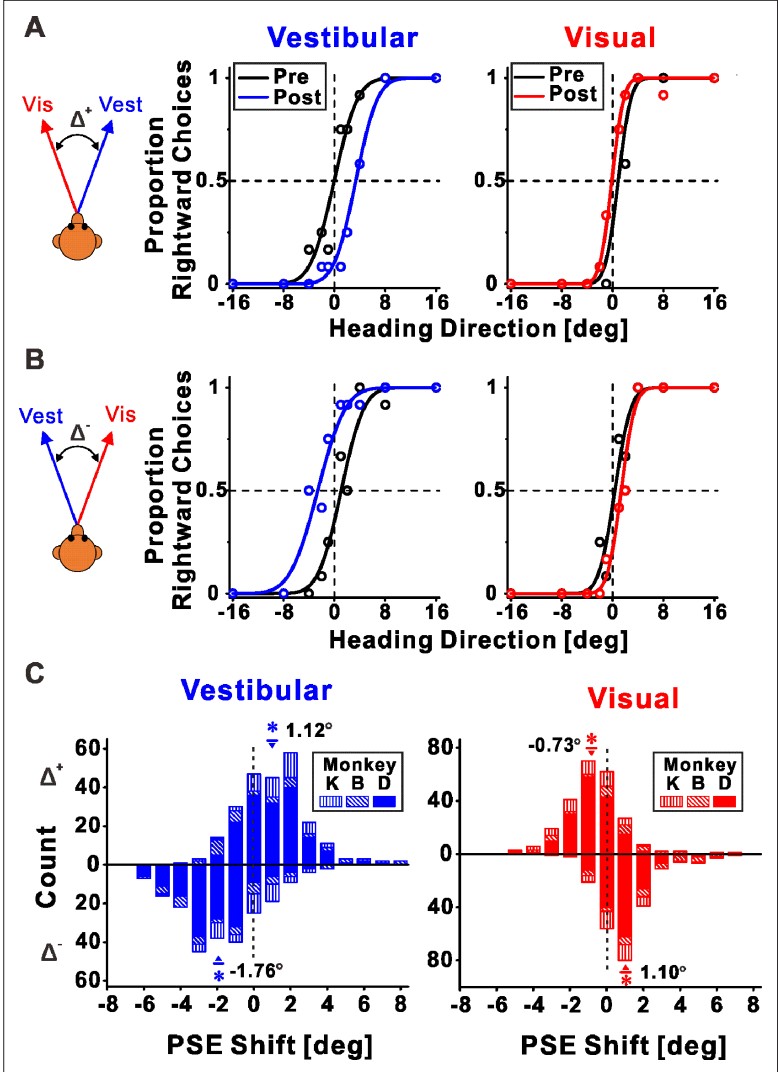

**Figure 2.** Multisensory recalibration behavior. (**A, B**) Perceptual recalibration in two example sessions, with (**A**) $\Delta^+$ (vestibular and visual headings offset to the right and left, respectively; monkey D, session #10) and (**B**) $\Delta^-$ (vestibular and visual headings offset to the left and right, respectively; monkey D, session #48). Psychometric curves (cumulative Gaussian distribution functions) were fitted to the data (circles), and represent the proportion of rightward choices, as a function of stimulus heading direction. Pre-recalibration heading judgments are depicted by black curves, in the left and right columns for vestibular and visual cues, respectively. After recalibration, vestibular and visual curves (blue and red, respectively) were shifted in relation to the pre-calibration curves. (**C**) Blue and red histograms represent the distributions of the point of subjective equality (PSE) shifts (post-recalibration minus pre-recalibration PSE) for vestibular and visual cues, respectively. Histograms above and below the abscissa represent sessions with $\Delta^+$ and $\Delta^-$, respectively. Inverted triangles (▼) and upright triangles (▲) with error bars represent mean ± standard error of the mean (SEM) shifts for sessions with $\Delta^+$ and $\Delta^-$, respectively. The numbers on the plots represent the mean PSE shifts. Asterisk symbols indicate significant shifts (p < 0.05). For the vestibular cue, p = $2.3 \times 10^{-17}$, N = 241 sessions ($\Delta^+$ condition), and p = $1.0 \times 10^{-28}$, N = 227 sessions ($\Delta^-$ condition), paired *t*-test. For the visual cue, p = $6.5 \times 10^{-11}$ ($\Delta^+$ condition), and p = $5.4 \times 10^{-23}$ ($\Delta^-$ condition), paired *t*-test. Summary statistics for the individual animals are presented in *Figure 2—source data 1*.

The online version of this article includes the following source data for figure 2:

**Source data 1.** Individual monkey summary statistics of behavioral shifts.

visual shifts (1.43° ± 0.09° and 0.91° ± 0.07°, respectively; p = $6.8 \times 10^{-5}$, paired *t*-test). This result is also consistent with our previous study. Thus, the behavioral results from the original study (performed in the Angelaki laboratory) were replicated in these experiments (in the Chen laboratory) using a new set of monkeys, with simultaneous neuronal recording. In the following sections, we present how

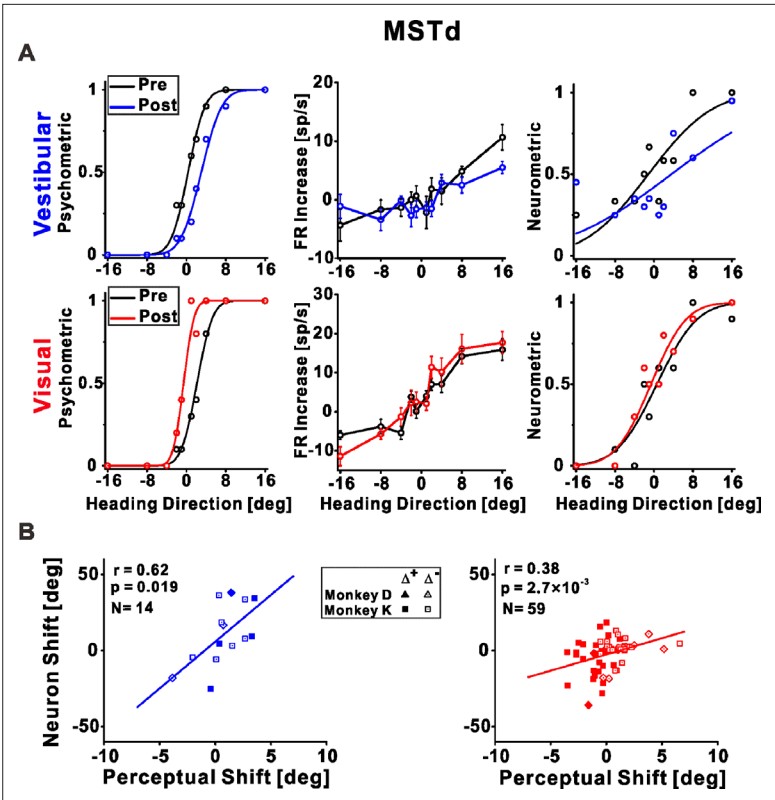

**Figure 3.** Dorsal medial superior temporal (MSTd) neuronal recalibration. (**A**) An example recalibration session (Δ⁺) with simultaneous recording from MSTd. The left column depicts the behavioral responses, pre-, and post-recalibration. The vestibular psychometric curve shifted 3.01° (to the right) and the visual curve shifted −2.71° (to the left). Neuronal responses (middle column) as a function of heading (pre- and post-recalibration). Circles and error bars represent average firing rates (FRs, baseline subtracted) ± standard error of the mean (SEM). The right column shows corresponding neurometric curves with fitted cumulative Gaussian functions. Each data point shows the proportion of trials in which an ideal observer would make a rightward choice given the FRs of the neurons. The vestibular neuronal shift was 4.73° (to the right) and the visual neuronal shift was −1.22° (to the left). (**B**) Correlations between neuronal point of subjective equality (PSE) shifts and perceptual PSE shifts for the vestibular and visual cues (left and right plots, respectively). Only neurons that passed screening (had significant responses and reliable neurometric PSEs, see Methods for details) were included in this analysis. Solid symbols represent sessions with Δ⁺ and open symbols represent Δ⁻. The solid lines illustrate the regression lines of the data. r, Pearson's correlation coefficient. Summary statistics for the individual animals, and linear mixed model (LMM) results, are presented in *Figure 3—source data 1* and *Figure 3—source data 2*, respectively.

The online version of this article includes the following source data and figure supplement(s) for figure 3:

**Source data 1.** Individual monkey summary statistics for dorsal medial superior temporal (MSTd) correlations.

**Source data 2.** Comparison of pooled model (PM) and linear mixed model (LMM) for dorsal medial superior temporal (MSTd).

**Figure supplement 1.** Distribution of neurometric point of subjective equality (PSE) reliability.

neuronal responses in areas MSTd, PIVC, and VIP (*Figures 3*, *4* and *5*, respectively) were recalibrated in comparison to the perceptual shifts.

## Vestibular and visual tuning in MSTd shifted according to their respective perceptual shifts

Responses of an example neuron recorded from MSTd during unsupervised recalibration are presented in *Figure 3A*. Behaviorally, the vestibular PSE shifted rightward and the visual PSE shifted leftward (left column, *Figure 3A*). Shifts in neuronal tuning could be subtle, therefore we used neurometrics to expose and quantify the neuronal shifts. Specifically, we calculated neurometric responses for the heading stimuli using the neuron's firing rates (FRs), and fit these with a cumulative Gaussian function.

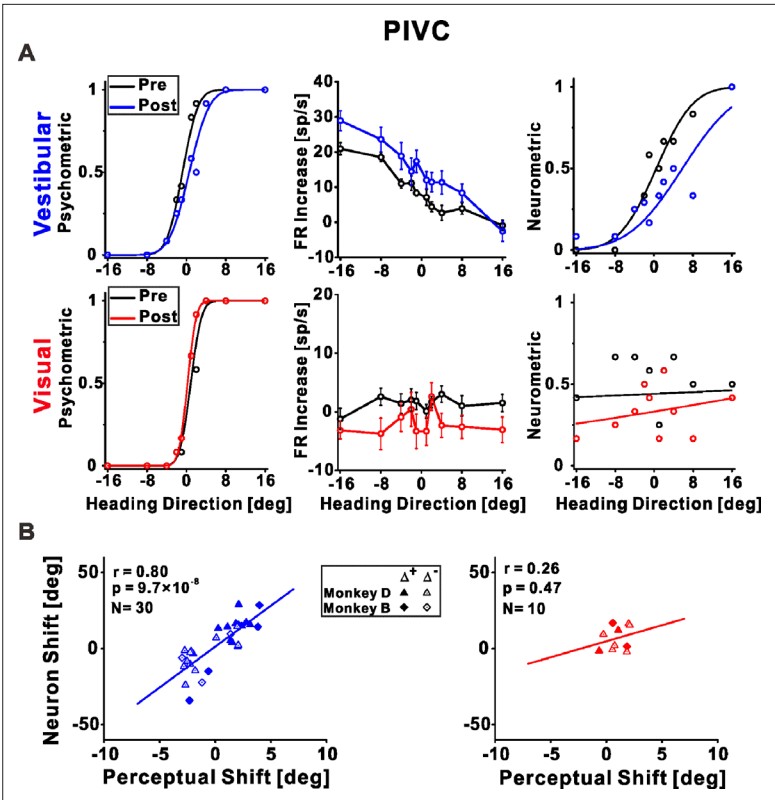

**Figure 4.** Parietoinsular vestibular cortex (PIVC) neuronal recalibration. (**A**) An example recalibration session (Δ⁺) with simultaneous recording from PIVC (conventions are the same as *Figure 3*). The vestibular and visual psychometric curves shifted 1.37° and −0.51° (to the right and left, respectively). The vestibular neurometric curve shifted 5.37° (to the right). Although a visual neurometric curve is presented for this example, no visual neurometric shift was calculated, and the neuron was excluded from subsequent visual cue analyses, because it did not pass the screening for significant tuning to visual stimuli. (**B**) Correlations between neuronal point of subjective equality (PSE) shifts and perceptual PSE shifts for the vestibular and visual cues. Summary statistics for the individual animals, and linear mixed model (LMM) results, are presented in *Figure 4—source data 1* and *Figure 4—source data 2*, respectively.

The online version of this article includes the following source data for figure 4:

**Source data 1.** Individual monkey summary statistics for parietoinsular vestibular cortex (PIVC) correlations.

**Source data 2.** Comparison of pooled model (PM) and linear mixed model (LMM) for parietoinsular vestibular cortex (PIVC).

The neurometric PSE reflects the heading direction at which the fitted neurometric curve crosses 0.5, that is, estimated straight ahead according to the neuronal data, in reference to the mean pre-recalibration FRs (see Methods for details). Neurometric curves for this example neuron are presented in the rightmost column of *Figure 3A*.

For this MSTd neuron, the vestibular neurometric curve shifted to the right, while the visual neurometric curve shifted to the left. Thus, the shifts in vestibular and visual tuning were consistent with the perceptual shifts. For subsequent (group) analyses, only neurons that both: (1) were significantly tuned to the respective (visual or vestibular) stimulus, and (2) had reliable neurometric PSEs, were included (see Methods and *Figure 3—figure supplement 1* for details). Neuronal shifts were calculated, similar to perceptual shifts, by the difference between the post- vs. pre-recalibration neurometric curves' PSEs. The MSTd neuronal shifts were significantly correlated with the perceptual shifts, both for vestibular and visual cues ($r = 0.62$, $p = 0.019$, $N = 14$, and $r = 0.38$, $p = 2.7 \times 10^{-3}$, $N = 59$, respectively; Pearson correlations, data pooled across monkeys). Similar results were found when analyzing the monkeys individually (*Figure 3—source data 1*) and when using a linear mixed model (LMM) which took into account differences between individual monkeys (the LMM did not provide a

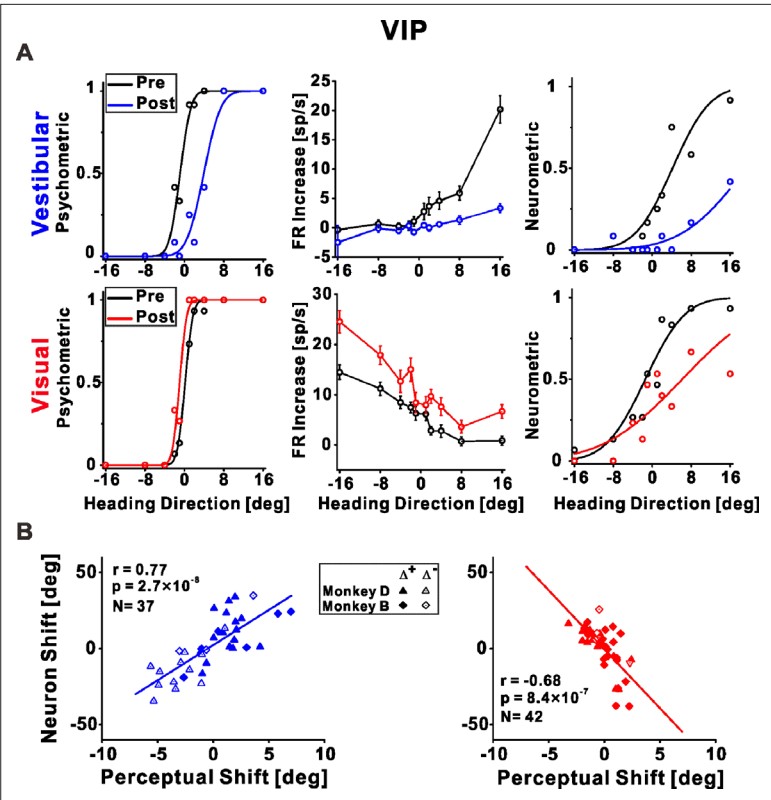

**Figure 5.** Ventral intraparietal (VIP) neuronal recalibration. (**A**) An example recalibration session (Δ⁺) with simultaneous recording from VIP (conventions are the same as **Figure 3**). The vestibular and visual psychometric curves shifted 4.81° and −1.13° (to the right and left, respectively). The vestibular and visual neurometric curves shifted 15.18° and 7.58°, respectively (both to the right). (**B**) Correlations between neuronal point of subjective equality (PSE) shifts and perceptual PSE shifts for the vestibular and visual cues. Summary statistics for the individual animals, and linear mixed model (LMM) results, are presented in **Figure 5—source data 1** and **Figure 5—source data 2**, respectively.

The online version of this article includes the following source data and figure supplement(s) for figure 5:

**Source data 1.** Individual monkey summary statistics for ventral intraparietal (VIP) correlations.

**Source data 2.** Comparison of pooled model (PM) and linear mixed model (LMM) for ventral intraparietal (VIP).

**Figure supplement 1.** Neuronal vs. behavioral shifts by neuron type in area ventral intraparietal (VIP).

better fit vs. the pooled model; **Figure 3—source data 2**). Therefore, in area MSTd neuronal recalibration occurs in accordance with perceptual recalibration, both for vestibular and visual cues.

## Vestibular tuning in PIVC shifted in accordance with vestibular perceptual shifts

In PIVC, a similar result was observed for vestibular tuning. The example vestibular neurometric curve (**Figure 4A**, top right) shifted to the right, which was consistent with the vestibular perceptual shift (**Figure 4A**, top left). Across the population of PIVC neurons, a significant positive correlation was seen between the neuronal and perceptual shifts for the vestibular cue ($r = 0.80$, $p = 9.7 \times 10^{-6}$, $N = 30$, Pearson correlation, data pooled across monkeys; **Figure 4B**, left panel). Similar results were found when analyzing the monkeys individually (**Figure 4—source data 1**) and when using an LMM (the LMM did not provide a better fit vs. the pooled model; **Figure 4—source data 2**).

In general, the PIVC neurons did not demonstrate robust responses to the visual stimuli. This example neuron was not significantly tuned to the visual stimuli (**Figure 4A**, bottom, middle), thus it had poor visual neurometric curves (**Figure 4A**, bottom, right) and was excluded from further visual (group) analyses. The correlation between the neuronal and perceptual shifts (performed for those neurons that did pass screening) was not significant for the visual cue ($r = 0.26$, $p = 0.47$, $N = 10$,

Pearson correlation, data pooled across monkeys). Similar results were found when analyzing the monkeys individually (*Figure 4—source data 1*) and when using an LMM (the LMM did not provide a better fit vs. the pooled model; *Figure 4—source data 2*). A Bayesian Pearson correlation ($BF_{10}$ = 0.49) supported neither the alternative hypothesis ($H_1$) of a correlation between neuronal and perceptual shifts for the visual cue, nor the null hypothesis ($H_0$). The lack of support for or against visual recalibration in PIVC primarily reflects the lack of robust tuning to visual heading stimuli in PIVC.

## Neuronal tuning in VIP to both vestibular and visual stimuli shifted according to vestibular perceptual shifts

*Figure 5A* presents an example neuron from VIP. The vestibular neurometric curve shifted rightward (*Figure 5A*, top right), in accordance with the vestibular perceptual shift (*Figure 5A*, top left). Surprisingly, the visual neurometric curve also shifted rightward (*Figure 5A*, bottom right). This was unexpected because the visual psychometric curve shifted leftward (*Figure 5A*, bottom left). Thus, while the vestibular and visual behavioral psychometric curves shifted in opposite directions (toward each other) the vestibular and visual neurometric curves shifted together, in accordance with the vestibular (not visual) perceptual shift.

Across the population of VIP neurons, the vestibular neurometric shifts were significantly positively correlated with the vestibular perceptual shifts ($r$ = 0.77, p = 2.7 × $10^{-8}$, N = 37, Pearson correlation, data pooled across monkeys; *Figure 5B*, left). Similar results were found when analyzing the monkeys individually (*Figure 5—source data 1*) and when using an LMM (the LMM did not provide a better fit vs. the pooled model; *Figure 5—source data 2*). Like in MSTd and PIVC, the positive correlation coefficient indicates that neuronal and behavioral curves shifted in the same direction for the vestibular cue.

By contrast, the visual neurometrics in VIP shifted in the opposite direction to the visual perceptual shifts. Neuronal and perceptual shifts for the visual cue were negatively correlated ($r$ = −0.68, p = 8.4 × $10^{-7}$, N = 42, Pearson correlation, data pooled across monkeys; *Figure 5B*, right). Similar results were found when analyzing the monkeys individually (*Figure 5—source data 1*) and when using an LMM (the LMM did not provide a better fit vs. the pooled model; *Figure 5—source data 2*). This exposes a striking mismatch between visual neuronal responses in VIP and visual perceptual function. It also exposes a striking mismatch between visual tuning in MSTd (which shifted in the same direction as visual perception) in comparison to visual tuning in area VIP (which shifted contrary to visual perception).

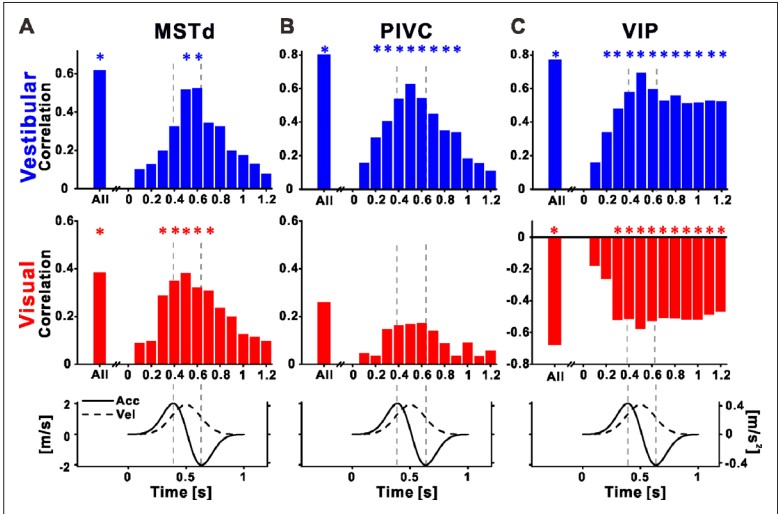

**Figure 6.** Recalibration of neuronal responses within the stimulus time course. Pearson correlations between neuronal and perceptual point of subjective equality (PSE) shifts, using the neuronal activity at specific time points during the stimulus, for (**A**) dorsal medial superior temporal (MSTd), (**B**) parietoinsular vestibular cortex (PIVC), and (**C**) ventral intraparietal (VIP). Top row: vestibular (blue histograms), middle row: visual (red histograms), bottom row: stimulus (acceleration and velocity) time course. Vertical dashed lines mark peak acceleration and peak deceleration. '*' symbols mark significant correlations.

To test whether this mismatch between behavior and tuning for visual cues in VIP relates to specific subtypes of neurons, we sorted the VIP data into three subsets: multisensory neurons (respond significantly to both vestibular and visual stimuli), and two groups of unisensory neurons (respond significantly exclusively to vestibular or visual stimuli). Similar results were seen for both multisensory and unisensory neurons (the neuronal–perceptual correlations remained consistently positive and negative for vestibular and visual cues, respectively; *Figure 5—figure supplement 1A*). We further sorted the multisensory neurons into those with congruent and opposite vestibular and visual heading preferences (*Chen et al., 2011a*; *Gu et al., 2006*) with no observable differences (*Figure 5—figure supplement 1B*). Therefore, the contrary shifts of visual tuning in VIP seem to reflect a general feature of this cortical area, rather than an anomaly of a subgroup of neurons.

## Temporal evolution of the correlation between neuronal and perceptual shifts

The neurometric curves in *Figures 3–5* were calculated using mean FRs averaged across the stimulus duration. But the self-motion stimuli generated by the platform and optic flow followed a specific dynamic time course, specifically, a Gaussian velocity profile and correspondingly a biphasic acceleration profile (see bottom row, *Figure 6*). Therefore, we further examined whether the correlations between neurometric and perceptual shifts depend on the time point within the stimulus interval.

For MSTd neurons, positive correlations (between neuronal and perceptual shifts) were seen for both vestibular and visual cues during the stimulus (*Figure 6A*). Correlations increased toward the middle of the stimulus, and dropped off rapidly at the end of the stimulus. Significant correlations (blue and red asterisk markers for vestibular and visual cues, respectively) were only seen around the middle of the stimulus. Thus neural recalibration in MSTd (in accordance with behavioral recalibration) could reflect the velocity responses.

For PIVC neurons, positive correlations (between neuronal and perceptual shifts) were seen only for vestibular cues, during the stimulus (upper panel in *Figure 6B*). Like MSTd, the vestibular correlations seemed to follow the velocity profile of the stimulus, with significant values around the middle of the stimulus. Correlations in the visual condition were very weak and not significant (middle panel in *Figure 6B*).

A very different profile was seen in VIP. Firstly, as described above, correlations between neuronal and perceptual recalibration were positive for the vestibular cue (upper panel in *Figure 6C*) and negative for the visual cue (middle panel in *Figure 6C*). Furthermore, the time course of these correlations was different in VIP: they increased in size gradually (positively for vestibular and negatively for visual), reaching a maximum around the middle of the stimulus epoch (the velocity peak), but significant correlations were found for time intervals beyond the end of the stimulus. This pattern is in line with sustained neuronal activity described previously for VIP. However, here this sustained activity correlated with subsequent vestibular choices, and was contrary to visual choices.

## VIP choice signals are reduced after cross-modal recalibration

Previous studies have found that neuronal responses in VIP are strongly influenced (sometimes even dominated) by choice signals (*Chen et al., 2021*; *Zaidel et al., 2017*). Hence our finding here, that neuronal tuning recalibrated contrary to perceptual shifts for the visual cue, was surprising and counterintuitive. We, therefore, wondered what happened to the strong choice signals for which VIP is renowned, which would predict that neuronal tuning (also for visual cues) would shift with behavior.

To visualize choice tuning for an example VIP neuron, we plotted 'choice-conditioned' tuning curves, namely, neuronal responses as a function of heading, separately for rightward and leftward choices (*Figure 7*). In the pre-recalibration block vestibular responses were strongly choice related (*Figure 7A*, left panel) – neuronal responses to the same heading stimulus were larger when followed by rightward (►, blue) vs. leftward (◄, cyan) choices (the blue line lies above the cyan line). After recalibration, the choice effect decreased (*Figure 7A*, right panel) – the choice-conditioned tuning curves were no longer separate. Similarly, visual responses were strongly choice-related pre-recalibration, and this decreased post-recalibration (*Figure 7B*). To quantify the choice (and sensory) components of neuronal activity, and to observe how these changed after recalibration, we applied a partial correlation analysis (*Zaidel et al., 2017*). For this example neuron, the partial choice correlation values ($R_c$, presented on the plots) were reduced both for vestibular and visual cues.

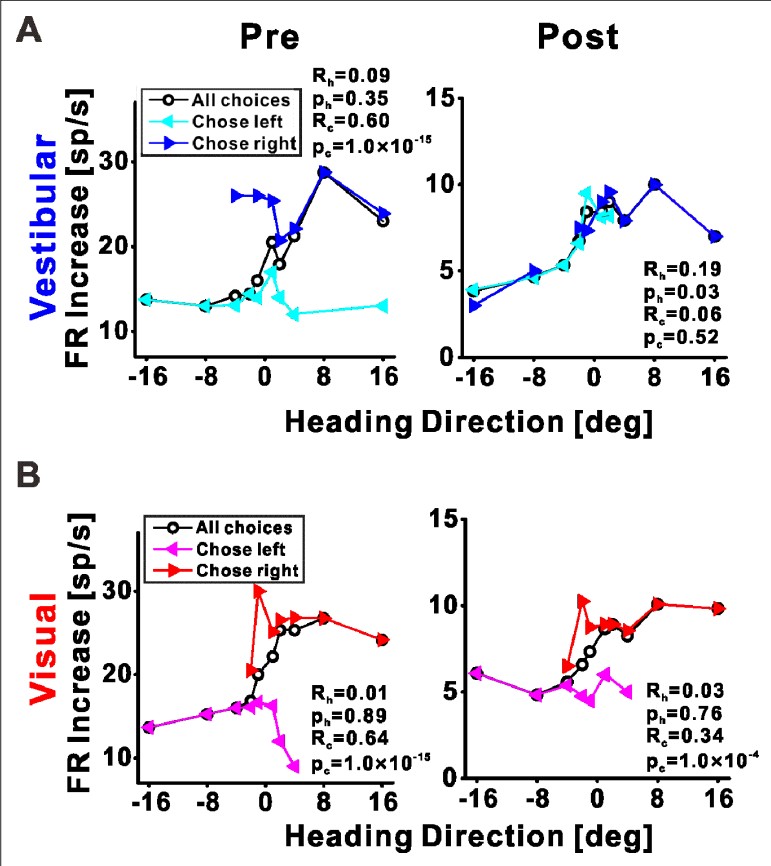

**Figure 7.** Choice tuning is reduced post-recalibration in an example ventral intraparietal (VIP) neuron. Neuronal responses for an example VIP neuron to (**A**) vestibular and (**B**) visual heading stimuli, pre- and post-recalibration (left and right columns, respectively). Blue and cyan curves depict choice-conditioned tuning curves (neuronal responses followed by rightward and leftward choices, respectively) for the vestibular cue. Red and magenta curves depict choice-conditioned tuning curves for the visual cue. Black curves (in the corresponding plots) represent all responses (not sorted by choice). Partial heading ($R_h$) and partial choice ($R_c$) correlations (with corresponding p values) are presented on the plots.

Across our sample of VIP neurons, the choice partial correlations in the post-recalibration block were significantly reduced compared to the pre-recalibration block, for both vestibular and visual cues (p = $6.0 \times 10^{-4}$ and p = $1.3 \times 10^{-3}$, respectively, paired *t*-tests; *Figure 8B*). However, the heading partial correlations ($R_h$) did not differ significantly from pre- to post-recalibration, neither for vestibular not visual cues (p = 0.96 and p = 0.85, respectively, paired *t*-tests; *Figure 8A*). For these statistical comparisons and for plotting we used the squared partial correlations (which quantify the amount of unique variance explained by choice or heading). We did not observe any significant changes in partial correlations in areas PIVC and MSTd (*Figure 8—figure supplement 1*). Lastly, there was no evidence for differences between post- and pre-recalibration baseline FRs in any of the three areas (*Figure 8— figure supplement 2*). Thus, shifts in neuronal tuning are not explained by changes in baseline activity.

## Discussion

This study provides the first demonstration of unsupervised (cross-modal) neuronal recalibration, in conjunction with perceptual recalibration, in single sessions. Single neurons from MSTd, PIVC, and VIP revealed clear but different patterns of recalibration. In MSTd, neuronal responses to vestibular and visual cues shifted – each according to their respective cues' perceptual shifts. In PIVC, vestibular tuning similarly shifted in the same direction as vestibular perceptual shifts (the PIVC cells were not robustly tuned to visual stimuli). However, recalibration in VIP was notably different: both vestibular and visual neuronal tuning shifted in the direction of the vestibular perceptual shifts. Thus, visual

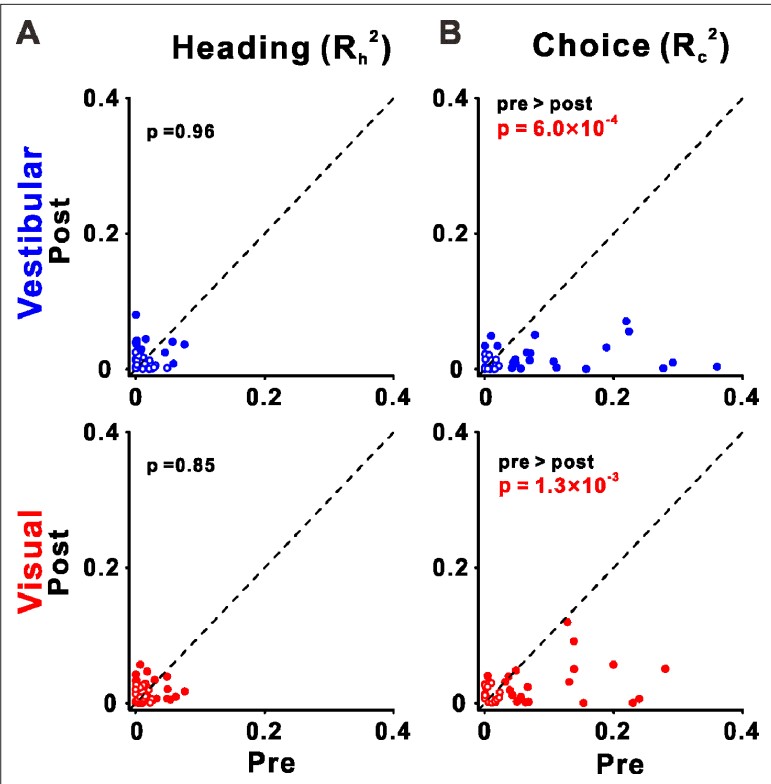

**Figure 8.** Choice tuning is reduced in ventral intraparietal (VIP) post-recalibration. (**A**) Heading and (**B**) choice partial correlation coefficients (squared) are depicted post- vs. pre-recalibration. Blue and red circles (top and bottom rows) represent vestibular and visual cues, respectively. Filled (empty) circles indicate significant (non-significant) partial correlations for heading or choice. p values are presented on the corresponding plots (two-tailed paired *t*-tests).

The online version of this article includes the following figure supplement(s) for figure 8:

**Figure supplement 1.** Choice and heading partial correlations.

**Figure supplement 2.** Baseline firing rates.

neuronal tuning shifted, surprisingly, contrary to visual perceptual shifts. These results indicate that neuronal recalibration differs profoundly across multisensory cortical areas.

## Neural correlates of vestibular–visual recalibration

To investigate the neuronal bases of unsupervised cross-modal recalibration, we first replicated the perceptual results from our previous study (*Zaidel et al., 2011*). Indeed, in the presence of a systematic vestibular–visual heading offset (with no external feedback) vestibular and visual cues both shifted in the direction required to reduce the cue conflict. And, as before, the vestibular shifts were larger compared to the visual shifts. Thus we confirmed robust recalibration of vestibular and visual cues, resulting from a systematic discrepancy between the cues' headings in an unsupervised context (i.e., without external feedback).

Since there was no external feedback regarding which cue was (in)accurate, unsupervised recalibration is driven by the cue conflict, presumably through an internal mechanism to maintain consistency between vestibular and visual perceptual estimates (*Zaidel et al., 2011*). Accordingly, we expected to see neuronal correlates of perceptual recalibration in early multisensory areas related to self-motion perception (*Zierul et al., 2017*), specifically: MSTd, which primarily responds to visual (but also vestibular) self-motion stimuli, and PIVC, which primarily responds to vestibular stimuli. We further expected that the neuronal recalibration in MSTd and PIVC would propagate to higher-level multisensory area VIP.

In MSTd, we indeed found that both visual and vestibular neuronal signals shifted, each in accordance with their corresponding cue's perceptual shifts. Hence, recalibration of visual self-motion responses was observed at least at the level of MSTd, which is the primary area in the visual hierarchy to respond to large field optic flow stimuli (*Britten, 2008*; *Britten and van Wezel, 1998*; *Britten and Van Wezel, 2002*; *Duffy and Wurtz, 1995*; *Gu et al., 2008*; *Gu et al., 2012*; *Gu et al., 2006*; *Wurtz and Duffy, 1992*). We cannot ascertain whether recalibration to visual responses occurred already in earlier visual regions, such as the middle temporal visual area, which projects to MSTd (*Maunsell and van Essen, 1983*; *Ungerleider and Desimone, 1986*), or whether it occurred only at the level of MSTd. Because MSTd is mainly a visual area, the recalibration of vestibular signals observed in MSTd likely occurred in upstream vestibular areas that project to MSTd, such as PIVC (*Chen et al., 2010*; *Chen et al., 2011a*). Indeed, robust vestibular recalibration (that was in line with the vestibular perceptual shifts) was observed in PIVC. Hence, neuronal correlates of perceptual recalibration were observed in relatively early multisensory areas related to self-motion perception (MSTd and PIVC).

## Modality-specific recalibration of vestibular and visual cues

Results from this experiment exposed modality-specific neuronal recalibration (in MSTd and PIVC). Namely, visual and vestibular tuning curves shifted differently (in opposite directions). This provides neuronal evidence against 'visual dominance', even for short-term recalibration (in single sessions). Rather, it supports the idea that cross-modal neuronal recalibration occurs also for visual (and not only for non-visual) cues. Furthermore, it exposes neuronal mechanisms to maintain internal consistency between vestibular and visual cues. This dynamic cross-modal plasticity may underlie our adept ability to adapt to sensory conflict commonly experienced in many modes of transport (on land, at sea, or in flight).

In a recent set of complementary studies, we tested *supervised* self-motion recalibration, by providing external feedback regarding cue accuracy (*Zaidel et al., 2021*; *Zaidel et al., 2013*). There, we found that supervised recalibration is a high-level cognitive process that compares the combined-cue (multisensory) estimate to feedback from the environment. Behaviorally, this resulted in 'yoked' recalibration – both cues shifted in the same direction, to reduce conflict between the combined estimate and external feedback (*Zaidel et al., 2013*). Neuronally, robust recalibration of both vestibular and visual neuronal tuning was seen in VIP, such that tuning for both cues shifted together, in accordance with the behavior (*Zaidel et al., 2021*).

However, because the shifts for both vestibular and visual cues were in the same direction in the supervised recalibration studies, neuronal tuning was also expected to shift in the same direction for both cues. Thus, we could not dissociate there whether neuronal shifts for a particular cue (e.g., visual) indeed followed the behavioral shifts for that cue (visual) or, less intuitively, the other cue (vestibular). By contrast, the unsupervised paradigm, tested in this study, elicits visual and vestibular shifts in opposite directions. It could thereby expose the (unexpected) finding that visual tuning in VIP actually shifts with vestibular (rather than visual) behavioral shifts.

The results here therefore also shed new light on the neuronal shifts observed in VIP after supervised recalibration (*Zaidel et al., 2021*). They indicate that yoking of visual and vestibular tuning is observed in VIP irrespective of the paradigm (supervised or unsupervised). Hence, yoked recalibration may be a feature of VIP, not just a feature of supervised recalibration.

## Contrary recalibration in higher-level area VIP

VIP is a higher-level multisensory area (*Bremmer et al., 2002*; *Colby et al., 1993*; *Duhamel et al., 1998*; *Schlack et al., 2002*; *Schlack et al., 2005*; *Schroeder and Foxe, 2002*) with clear vestibular and visual heading selectivity (*Chen et al., 2011a*; *Chen et al., 2011b*). But the nature of these self-motion signals in VIP is not fully understood. In contrast to our prediction that recalibrated signals in MSTd and PIVC would simply propagate to VIP, we found a different and unexpected pattern of recalibration in VIP. While vestibular tuning shifted in line with vestibular perceptual shifts (like MSTd and PIVC), visual tuning shifted opposite in direction to the visual perceptual shifts (and opposite in direction to MSTd visual recalibration). These findings indicate that visual responses in VIP do not reflect a simple feed-forward projection from MSTd. They also suggest that visual responses in VIP are not decoded for heading perception (otherwise these would not shift in opposite directions). This interpretation is in line with findings that inactivation (*Chen et al., 2016*) and microstimulation (*Yu and*

*Gu, 2018*) in VIP do not affect perceptual decisions. Thus, the convergence of visual and vestibular signals in VIP likely serves purposes other than cue integration.

We previously found strong choice-related activity in VIP neurons (*Zaidel et al., 2017*). Accordingly, we considered that shifts in VIP neuronal tuning (after supervised recalibration) might simply reflect the altered choices (*Zaidel et al., 2021*). However, choice-related activity cannot explain the results here, because the predicted shifts in neuronal tuning would be in the same direction as the altered choices (perceptual shifts), whereas we found contrary visual recalibration. To understand contrary shifts that could arise despite strong choice-related activity in VIP, we investigated choice tuning pre- and post-recalibration in VIP neurons. We found that choice tuning in VIP decreased after unsupervised recalibration. This allowed contrary shifts to be exposed, and opens up new and fascinating questions regarding the purpose of contrary visual recalibration in VIP.

Because visual and vestibular tuning in VIP both shifted in the same direction (in accordance with vestibular perceptual shifts) we speculate that VIP recalibration reflects a global shift in the vestibular reference frame. This notion is consistent with suggestions that VIP encodes self-motion and tactile stimuli in head or body-centered coordinates (*Avillac et al., 2005*; *Avillac et al., 2004*; *Chen et al., 2013b*; *Chen et al., 2018*; *Zhang et al., 2004*), and that visual signals in VIP are remapped within these coordinates (*Avillac et al., 2005*; *Sereno and Huang, 2014*). Accordingly, visual responses in VIP are transformed into a vestibular-recalibrated space. This leads to a remarkable dissociation between visual tuning in VIP and MSTd. Interestingly, visual self-motion perception follows the MSTd (not VIP) recalibration. This is in line with a causal connection between MSTd and visual heading discrimination (*Britten and van Wezel, 1998*; *Gu et al., 2012*).

What purpose might such visual signals in VIP serve? One possible idea is that they might reflect an expectation signal – for example, predicted vestibular or somatosensory sensation, based on the current visual signal. During combined stimuli (in the recalibration and post-recalibration blocks), the visual signal always appeared together with the vestibular sensory input. Thus, if visual responses in VIP reflect vestibular expectations, then these would shift together with vestibular (rather than visual) recalibration.

## Limitations and future directions

Our results revealed correlations between neuronal recalibration and perceptual recalibration. However, they do not implicate any causal connections. Therefore, whether these cortical areas are actively involved in cross-modal recalibration (i.e., play a causal role) vs. simply reflecting the recalibrated signals (without playing a causal role) requires further research. To probe more directly for causal links, direct manipulation of neuronal activity might be required. For example, would reversible inactivation or microstimulation (of one or a combination of these multisensory areas) eliminate (or bias) unsupervised recalibration? In addition, future studies are needed to examine how the systematic error between vestibular and visual heading signals is detected. This likely involves additional brain areas, for example, the cerebellum, implicated in internal-model-based error monitoring (*Markov et al., 2021*; *Rondi-Reig et al., 2014*), and/or the anterior cingulate cortex, implicated in conflict monitoring (*Bush et al., 2000*; *Holroyd and Coles, 2002*). Thus, a wide-ranging effort to record and manipulate neural activity across a variety of brain regions will be necessary to tease apart the circuitry underlying this complex and important function.

The lack of evidence for (or against) visual recalibration in PIVC primarily reflects the lack of robust tuning to visual heading stimuli. We interpret the observed shifts in vestibular tuning in PIVC as lower-level, sensory, recalibration (similar to MSTd) based on the broader understanding that PIVC encodes lower-level vestibular signals, with transient time courses, and impoverished visual tuning (*Chen et al., 2016*; *Chen et al., 2021*). Our results are in line with this interpretation, and there is no reason to suspect that PIVC reflects more complex multisensory recalibration (like VIP). Nonetheless, the data could also be in line with alternative interpretations. A broader range of headings, and analyses beyond neurometrics, would be required to better understand whether (and how) visual signals in PIVC might be recalibrated.

The most surprising and intriguing finding in this study was the contrary recalibration of visual tuning in VIP. We propose that yoked recalibration of visual and vestibular responses in VIP (despite differential perceptual recalibration) might reflect a global shift in vestibular space. Accordingly, we suggest that visual responses in VIP might reflect an expectation signal (in vestibular space), for example, a

simulation of the expected corresponding vestibular response (or integrated position, because VIP responses are sustained beyond the stimulus period). However, this idea is speculative, and the data from this study cannot address this question. Hence, further research is needed to investigate this idea, for example, by conditioning expectations for vestibular motion on other (non-motion) cues, and investigating whether these cues can induce simulated vestibular responses. If this hypothesis turns out to be true, it could greatly contribute to our understanding regarding the functions of the parietal cortex, and the brain mechanisms of perceptual inference.

## Concluding remarks

This study exposed modality-specific recalibration of neuronal signals, resulting from a cross-modal (visual–vestibular) cue conflict. It further revealed profound differences in neuronal recalibration across multisensory cortical areas MSTd, PIVC, and VIP. The results therefore provide novel insights into adult multisensory plasticity, and deepen our understanding regarding the different functions of these multisensory cortical areas.

## Methods

### Subjects and surgery

Three male rhesus monkeys (*Macaca mulatta*, monkeys D, B, and K) weighing 8–10 kg participated in the experiment. The monkeys were first trained to sit in a custom primate chair and gradually exposed to the laboratory environment. Then the monkeys were chronically implanted a head-restraint cap and a sclera coil for measuring eye movement. After full recovery, the monkeys were trained to perform experimental tasks. All animal surgeries and experimental procedures were approved by the Institutional Animal Care and Use Committee at East China Normal University (IACUC protocol number: Mo20200101).

### Equipment setup and motion stimuli

During the experiments, the monkeys were head-fixed and seated in a primate chair which was secured to a six degrees of freedom motion platform (Moog, East Aurora, NY, USA; MB-E-6DOF/12/1000 kg). The chair was also inside a magnetic field coil frame (Crist Instrument Co, Inc, Hagerstown, MD, USA) mounted on the platform for measuring eye movement with the sclera coil technique (for details, see *Zhao et al., 2021*).

Vestibular stimuli corresponded to linear movements of the platform (for details, see *Chen et al., 2013a*; *Gu et al., 2006*; *Zhao et al., 2021*). Visual stimuli were presented on a large computer screen (Philips BDL4225E, Royal Philips, Amsterdam, Netherlands), attached to the field coil frame. The display (62.5 cm × 51.5 cm) was viewed from a distance of 43 cm, thus subtending a visual angle of 72° × 62°. The sides of the coil frame were covered with a black enclosure, so the monkey could only see the visual stimuli on the screen (*Gu et al., 2006*; *Zhao et al., 2021*). The display had a pixel resolution of 1920 × 1080 and was updated at 60 Hz. Visual stimuli were programmed in OpenGL to simulate self-motion through a 3D cloud of 'stars' that occupied a virtual cube space 80 cm wide, 80 cm tall, and 80 cm deep, centered on the central fixation point on the screen. The 'star' density was 0.01/cm$^3$. Each 'star' comprised a triangle with base by height: 0.15 cm × 0.15 cm. Monkeys wore custom stereo glasses made from Wratten filters (red #29 and green #61; Barrington, NJ, USA), such that the optic flow stimuli could be rendered in three dimensions as red-green anaglyphs.

The self-motion stimulus was either vestibular-only, visual-only, or combined (visual and vestibular stimuli). In the vestibular-only condition, there was no optic flow on the screen and the monkey was translated by the motion platform. In the visual-only condition, the motion platform remained stationary while optic flow was presented on the screen. For the combined condition, the monkeys experienced both translation and optic flow simultaneously. Each motion stimulus followed a Gaussian velocity profile with a duration of 1 s, and a displacement amplitude of 13 cm (bottom row, *Figure 6*). The peak velocity was 0.41 m/s, and the peak acceleration was 2.0 m/s$^2$.

### Task and recalibration protocol

The monkeys were trained to report their perceived direction of self-motion in a two-alternative forced-choice (2AFC) heading discrimination task (for details, see *Chen et al., 2013a*; *Gu et al.,*

*2008*). In each trial, the monkey primarily experienced a forward motion with a small leftward or rightward component. During stimulation, the monkey was required to maintain fixation on a central point, within a 3° × 3° window. At the end of the trial (after a 300-ms delay period beyond the end of the stimulus), the monkeys needed to make a saccade to one of two targets (located 5° to the left and right of the central fixation point) to report their motion percept as leftward or rightward relative to straight ahead. The saccade endpoint had to remain within 2.5° of the target for at least 150 ms to be considered a valid choice. Correct responses were rewarded with a drop of liquid.

To elicit recalibration, we used an unsupervised cue-conflict recalibration protocol previously tested behaviorally in humans and monkeys (*Zaidel et al., 2011*). Each experimental session consisted of three consecutive blocks, as described below.

### Pre-recalibration block

This block was used to deduce the baseline performance (psychometric curve) of each modality for the monkeys, thus only a single-cue (vestibular-only or visual-only) stimulus was presented (*Figure 1A*). Across trials, the heading angle was varied in small steps around straight ahead. Ten logarithmically spaced heading angles were tested for each monkey (±16°, ±8°, ±4°, ±2°, and ±1°). To accustom the monkeys to not getting a reward for all the trials, they were rewarded with 95% probability for correct choices, and not rewarded for incorrect choices.

### Recalibration block

Only combined vestibular–visual cues were presented in this block (*Figure 1B*). A discrepancy (Δ) between the vestibular and visual cues was introduced gradually from 2° to 10° (or −2° to −10°) with steps of 2°, and then held at ±10° for the rest of the block. This gradual introduction was applied to avoid the monkeys from noticing the discrepancy. The sign of Δ represents the orientation of the discrepancy: positive Δ (marked by Δ⁺) indicates that the vestibular and visual cues were systematically offset to the right and to the left, respectively, and vice versa for negative Δ (Δ⁻). Only one discrepancy orientation (Δ⁺ or Δ⁻) was used per session. The combined stimulus headings followed the same ten headings as the single-cue stimuli in the pre-recalibration block. For the combined stimuli, the vestibular and visual headings were each offset by Δ/2 (to opposite sides), such that the combined heading was defined in the middle between the vestibular and visual headings. Unlike the pre-recalibration block, monkeys only needed to maintain fixation on the central fixation point during the stimulus presentation and did not need to make choices at the end of trials. They were rewarded for all the trials for which they maintained fixation. 7–10 repetitions were run for each Δ increment, and an additional 10–16 repetitions were run for maximum Δ (±10°).

### Post-recalibration block

During this block, performance for the individual (visual/vestibular) modalities was once again tested using single-cue trials (as in the pre-recalibration block). Responses to these trials were used to measure recalibration. The single-cue trials were interleaved with combined-cue trials (with a ±10° discrepancy, like the end of the recalibration block, *Figure 1C*). The combined-cue trials were interleaved to maintain recalibration while it was measured (for details, see *Zaidel et al., 2011*). To avoid perturbing the recalibrated behavior, we adjusted the reward probability for single-cue trials as follows: if the single-cue heading was of relatively large magnitude, such that, if it were part of a combined-cue trial also the other cue would lie to the same side (right or left), monkeys were rewarded as in the pre-recalibration block (95% probability reward for correct choices; no reward for incorrect choices). If, however, the heading for other modality would have been to the opposite side, the monkeys were rewarded stochastically (70% reward probability, regardless of their choices).

## Electrophysiological recordings

We recorded extracellular activity from isolated single neurons in areas MSTd, PIVC, and VIP using tungsten microelectrodes (Frederick Haer Company, Bowdoin, ME, USA; tip diameter ~3 µm; impedance, 1–2 MΩ at 1 kHz). The microelectrode was advanced into the cortex through a transdural guide tube, using a hydraulic microdrive (Frederick Haer Company). Raw neural signals were amplified, band-pass filtered (400–5000 Hz), and digitized at 25 kHz using the AlphaOmega system (AlphaOmega Instruments, Nazareth Illit, Israel). Spikes were sorted online, and spike times along with all

behavioral events were collected with 1-ms resolution using the Tempo system. If the online sorting was not adequate, offline spike sorting was performed.

The target areas (MSTd, PIVC, and VIP) were identified based on the patterns of gray and white matter transitions, magnetic resonance imaging scans, stereotaxic coordinates, and physiological response properties as described previously (MSTd: *Gu et al., 2006*; PIVC: *Chen et al., 2010*; VIP: *Chen et al., 2011a*).

## Data analysis

Data analysis was performed with custom scripts in Matlab R2016a (The MathWorks, Natick, MA, USA). Psychometric plots were constructed by fitting the proportion of 'rightward' choices as a function of heading angle with a cumulative Gaussian distribution function, using the *psignifit* toolbox for MATLAB (version 2.5.6). Separate psychometric functions were constructed for each cue (visual and vestibular) and block (pre- and post-recalibration). The psychophysical threshold and PSE were defined, respectively, by the standard deviation (SD, $\sigma$) and mean ($\mu$) of the fitted Gaussian function. The PSE represents the heading angle that would be perceived as straight ahead, also known as the 'bias'. Vestibular and visual recalibration (PSE shift) was calculated for each session by subtracting the pre-recalibration PSE from the post-recalibration PSE:

$$PSE\ shift = PSE_{post} - PSE_{pre} \tag{1}$$

Neuronal tuning curves were constructed for vestibular and visual cues, pre- and post-recalibration, by calculating the average (baseline subtracted) FR responses, as a function of heading. FR responses were calculated over the duration of stimulus presentation ($t = 0–1$ s), and baseline FRs were calculated (per block) by the average FR in the 1-s window before stimulus onset. A neuron was considered 'tuned' if the linear regression of FR responses by heading (over the narrow range of stimuli presented: $-16°$ to $16°$) had a significant slope ($p < 0.05$).

This selection criterion was selective for neurons that have sloped tuning around straight ahead, and excluded neurons with flat tuning, or a tuning preference, straight ahead. In the cortical areas of interest in this study, a disproportionately large number of neurons have steep tuning slopes around straight ahead (*Chen et al., 2011b*; *Gu et al., 2010*). These neurons are most informative for heading discrimination (large Fisher information, *Gu et al., 2010*). By contrast, neurons with relatively flat tuning around straight ahead are less informative for heading discrimination (low Fisher information). Accordingly, small shifts can be readily detected in neurons with sloped tuning (but not in those with flat tuning) around straight ahead. Therefore, in this study we focused on the prevalent neurons with sloped tuning around straight ahead.

## Neurometrics

For each neuron recorded, neurometric curves (per cue and block) were constructed from the FRs (*Chen et al., 2013a*; *Fetsch et al., 2011*; *Gu et al., 2008*; *Gu et al., 2007*). For this, the FRs were first normalized (*z*-scored) by subtracting the pre-recalibration mean, and dividing by the pre-recalibration SD. The same (pre-recalibration) mean and SD values were used to normalize both the pre- and post-recalibration FRs (per cue). A common reference (pre-recalibration mean, corresponding to *z*-score $= 0$) was needed to expose PSE shifts (calculating neurometric curves by comparing responses to positive vs. corresponding negative headings assumes PSE $= 0°$).

Then, for each heading, an ROC (receiver operating characteristic) curve was computed by moving a 'criterion' value from the minimum to the maximum *z*-score (in 100 steps), and plotting the probability that the *z*-scores exceeded the criterion vs. whether *z*-score $= 0$ (the pre-recalibration mean) exceeded that same criterion, or not (1 or 0, respectively). A single point on the ROC curve was produced for each increment in the criterion. The area under the ROC curve reflects the probability that an ideal observer would discriminate the neuronal responses for the given heading to the neuron's preferred (vs. non-preferred) side (right/left), in relation to the pre-recalibration mean. Then these values were mapped onto the probability of a rightward choice and fitted with a cumulative Gaussian function (similar to perceptual psychometrics).

## Neuronal shifts

For subsequent analyses, that is, calculating neurometric shifts (and comparison thereof to perceptual shifts) only neurons that passed both of the following two screening criteria (per cue) were included:

(1) significant tuning to the corresponding cue (either pre- or post-recalibration; see Data analysis subsection above for details). (2) Both the pre- and post-recalibration neurometrics produced reliable PSEs (bootstrapped SD of the PSE <10°, both pre- and post-recalibration). The bootstrapped SDs of the PSEs (for the neurons that passed the first criterion, of significant tuning) are presented in *Figure 3—figure supplement 1*. This resulted in 14 and 59 MSTd neurons for vestibular and visual cues, respectively (*Figure 3*); 30 and 10 PIVC neurons for vestibular and visual cues, respectively (*Figure 4*); 37 and 42 PIVC neurons for vestibular and visual cues, respectively (*Figure 5*).

Neuronal shifts were measured by the difference between the post- and pre-recalibration neurometric PSEs (similar to perceptual shifts, see *Equation. 1*). For each recording area (MSTd, PIVC, and VIP) and cue (vestibular and visual) neuronal shifts were compared to perceptual shifts, using Pearson correlations (pooling data across monkeys). Additionally, to assess the relationship between neuronal and perceptual shifts, while taking into account the differences of individual monkeys, we used an LMM, which allowed for random effects in slope and intercept for the different monkeys. The goodness of fit was assessed for the LMM and the pooled model (which did not take into account differences of individual monkeys) using AIC (Akaike Information Criterion) and BIC (Bayesian Information Criterion) (*Vrieze, 2012*). The LMM did not provide better fits vs. the pooled model, and the results (fixed effects) remained similar compared to the pooled model (*Figure 3—source data 2*, *Figure 4—source data 2*, and *Figure 5—source data 2*).

To measure neuronal shifts at different time points during the stimulus, we calculated neurometric shifts based on FRs in narrow (200 ms) windows, in increments of 100 ms. The time index (the center of the window) ranged from $t = 0.1$ s to $t = 1.2$ s (relative to stimulus onset). This range did not include the choice saccade, which could only take place after $t = 1.3$ s because of the delay period (300 ms) between the offset of the stimulus and the onset of the saccade targets. All neurons that passed both of the screening criteria (described above) were included in this analysis.

## Partial correlation analysis

To disassociate the unique contributions of heading stimuli and choices to the neural responses (FRs) we computed Pearson partial correlations between these variables (for details, see *Chen et al., 2021*; *Zaidel et al., 2017*). This produced: (1) a heading partial correlation ($R_h$) that captured the linear relationship between FRs and headings, given the monkey's choices, and (2) a choice partial correlation ($R_c$) that captured the linear relationship between FRs and choices, given the stimulus headings. Partial correlations were calculated based on data acquired over the entire stimulus duration. Positive (negative) heading partial correlations indicate that FRs were greater (smaller) for rightward vs. leftward headings (given the choices). Likewise, positive (negative) choice partial correlations indicate that FRs were greater (smaller) for rightward vs. leftward choices (given the stimulus headings).

## Statistical analysis

To evaluate differences in monkey behavior (PSE), heading, or choice partial correlations, between pre- and post-recalibration, we used two-tailed paired *t*-tests. Possible differences in spontaneous (baseline) FRs between pre- and post-recalibration were evaluated using Bayesian paired-samples *t*-tests ($BF_{10}$ values). Relationships between neuronal and perceptual shifts were tested using Pearson's correlation coefficients and LMMs. Statistical analysis was conducted using JASP (Version 0.16.3) and R (Version 4.2.2).

## Acknowledgements

This work was supported by grants from the 'STI2030-major projects' (No. 2021ZD0202600), the National Basic Research Program of China (No. 32171034) to AC, and the ISF-NSFC joint research program to AC (No. 32061143003) and AZ (No. 3318/20). We thank Prof. Dora Angelaki for the helpful comments. We are also grateful to Minhu Chen for outstanding computer programming.

# Additional information

## Funding

| Funder | Grant reference number | Author |
| --- | --- | --- |
| Ministry of Science and Technology of the People's Republic of China | 2021ZD0202600 | Aihua Chen |
| National Natural Science Foundation of China | 32171034 | Aihua Chen |
| National Natural Science Foundation of China | 32061143003 | Aihua Chen |
| Israel Science Foundation | 3318/20 | Adam Zaidel |

The funders had no role in study design, data collection, and interpretation, or the decision to submit the work for publication.

## Author contributions

Fu Zeng, Data curation, Formal analysis, Validation, Visualization, Methodology, Writing - original draft; Adam Zaidel, Conceptualization, Formal analysis, Supervision, Funding acquisition, Validation, Methodology, Writing - review and editing; Aihua Chen, Conceptualization, Data curation, Formal analysis, Supervision, Funding acquisition, Investigation, Methodology, Writing - original draft, Project administration, Writing - review and editing

## Author ORCIDs

Fu Zeng http://orcid.org/0009-0000-6857-6485
Adam Zaidel http://orcid.org/0000-0003-4405-8717
Aihua Chen http://orcid.org/0000-0001-5066-2844

## Ethics

All animal surgeries and experimental procedures were approved by the Institutional Animal Care and Use Committee at East China Normal University (IACUC protocol number: Mo20200101).

## Decision letter and Author response

Decision letter https://doi.org/10.7554/eLife.82895.sa1
Author response https://doi.org/10.7554/eLife.82895.sa2

# Additional files

## Supplementary files

• MDAR checklist

## Data availability

The data and analysis code for this study have been uploaded to Github and can be found at https://github.com/FuZengBio/Recalibration (copy archived at *Zeng, 2022*).

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
