## [Editor Report]

This important study combines quantitative behavior and single-unit recordings in nonhuman primates to investigate the role of three cortical areas in cross-modal sensory calibration, a form of neural plasticity that is important for perception and learning. The results convincingly demonstrate key similarities and striking differences across the three areas and provide the first evidence for this form of calibration (in correspondence with behavior) at the level of single neurons. The work will be of broad interest to neuroscientists and psychologists studying multisensory perception, plasticity, and the role of sensory and association cortices in perceptual decisions.

---

## [Decision Letter]

**Decision letter after peer review:**

Thank you for submitting your article "Contrary neuronal recalibration in different multisensory cortical areas" for consideration by *eLife*. Your article has been reviewed by 3 peer reviewers, and the evaluation has been overseen by a Reviewing Editor and Michael Frank as the Senior Editor. The following individuals involved in review of your submission have agreed to reveal their identity: Umberto Olcese (Reviewer #2); Robbe Goris (Reviewer #3).

The reviewers' assessment of the work is overall quite positive; congratulations on an interesting and potentially impactful study. The reviewers have discussed their reviews with one another, and the Reviewing Editor has drafted this to help you prepare a revised submission.

Essential revisions:

1) Please report summary statistics (means and correlation coefficients) and p values for individual animals, for the data in Figure 2C, 3B, 4B, and 5B. This is not to say that the results must be significant at the individual animal level in order to support the study's main conclusions; we are all aware of the practicalities and accepted conventions of the field. Yet we feel it is important to be up front about this limitation, if indeed some of the individual monkey results fail to reach significance.

To be clear, the reasoning behind this request is that repeated observations from the same subject/condition are not independent of each other, thus correlations calculated from pooled data might obscure, inflate, or even reverse the true relationship between the variables (e.g., Simpson's Paradox).

Apart from individual subject analyses, another approach (indeed our strong recommendation) is to perform a hierarchical analysis. There are multiple options for this; probably the easiest one is to calculate linear mixed regression models (LMM) with one variable as predictor, the other as outcome and no intercept. This can be done in the frequentist way using (for example) the lmer package in R, or in a Bayesian way using Stan, which even as an automatized module for GLMMs. Partial correlations can be achieved by adding the variable that is partialized out as a predictor.

In summary, the authors may choose one (or more) of the three mentioned approaches to enhance statistical rigor: (1) separate t-tests and correlations for each animal and condition, (2) frequentist hierarchical linear models, and (3) Bayesian hierarchical linear models. It is worth pointing out that Bayesian approaches have advantages over frequentist methods, for instance quantifying the evidence for the null hypothesis (and thus better evaluating negative results such as Figure 4B-right).

2) Please consider applying a goodness-of-fit criterion to the neurometric curves before inclusion of their PSEs in the neuron-behavior correlation analyses – AND/OR evaluate the reliability of the PSEs using standard error obtained from the fitting procedure, or a bootstrap-based confidence interval. We would not require individual neuron shifts (δ-PSE) to be significant according to such SEs (i.e., through error propagation), but a reanalysis after removing particularly poor fits seems appropriate. The criterion to use for this is difficult for us to specify and thus can use your best judgment.

3) Depending on how the first two points are handled, and the outcome thereof, it may be necessary to scrutinize the role of outliers in generating the correlation coefficients and p values obtained. For instance, is the correlation of Figure 3B-left still significant without the upper four points, and 3B-right without the rightmost three points? A hierarchical analysis and/or neurometric goodness-of-fit criterion could reduce the role of outliers, in which case no formal outlier correction or other way to address this is needed.

*Reviewer #1 (Recommendations for the authors):*

The results are really interesting, yet, the manuscript in its current form needs revisions along two dimensions, 1) data analysis and 2) writing.

Methods

I am aware that basically all analyses in this manuscript have been used in published papers. The problems outlined below hold for those papers, too, sorry. Bad (or good) luck having this time a reviewer from another field.

Correlations derived from data that includes multiple repetitions per subject (and condition) require a hierarchical analysis because repeated observations from the same observer are not independent of each other. Subject A might on average have higher shifts (behavioral and neurometric) than subject B. In that case, a non-hierarchical analysis might return a significant positive correlation even if within subject A negative behavioral shifts are associated with positive neurometric shifts and vice versa. There are multiple ways to account for intra-subject correlations, probably the easiest one is to calculate linear mixed regression models (LMM) with one variable as predictor, the other as outcome and no intercept. This can be done in the frequentist way using the lmer package in R and in a Bayesian way using Stan which even as an automatized module for GLMMs. Partial correlations can be achieved by adding the variable that is partialized out as a predictor. Of course, it is also possible to just analyze each subject separately.

The same holds for any comparisons of mean values (across heading offsets during the recalibration phase or across modalities). For example, a t-test that includes perceptual shifts from all sessions and all monkeys is not valid as the values from a single monkey are correlated. Again, there are several ways to account for this. For example, an LMM with cross-modal discrepancy as predictor (and a free intercept) or each monkey's data could be tested separately.

The procedure to derive neurometric curves, which are central to the study, should be explained better. In the main text and in the figures, it should be clarified that each data point shows the proportion of trials in which an ideal observer would make a rightwards choice given only the firing rates of the neurons (and assumed anti-neurons). In the methods section, it should be explained in detail how the ROC curve is derived. Readers unfamiliar with the method should understand that the ROC incorporates decisions for a range of decision thresholds and that the area under the ROC curve (AUROC, not ROC value) corresponds to a theoretical observer's ability to discriminate between leftwards and rightwards motion given firing rates across repetitions of a specific motion direction. Finally, those AUROC values (ranging from 0.5 to 1) are mapped onto the probability of a rightwards choice (ranging from 0 to 1) given a heading direction.

The use of PSEs for the correlational analysis should be conditional on the goodness of fit of the sigmoid and the SD should fall within a reasonable range. A PSE derived from an ill-fitting or very flat curve has no informational value and is likely to be extreme, which in turn is huge a problem for any correlational analysis.

The methods section states that only neurons tuned to heading directions as indicated by cues in a specific modality were included in the neurometric analysis for that modality. If that exclusion criterion was applied, why are the neurometric curves for 'the other modality' so bad? I would have expected that neurons tuned to visual heading cues result in a good neurometric function for visual heading no matter where these neurons are located in the brain. In turn, I would have expected flat neurometric functions for visual heading direction in PIVC if also neurons tuned to vestibular heading information were included in the analysis. What am I missing?

Another instance of me probably not getting something: to identify neurons tuned for heading directions as cued by one of the two modalities, spike rates were regressed onto heading directions and only those neurons with a significant slope were included. I get that the range of tested headings is smaller than the width of a typical tuning curve and thus a linear regression makes more sense than trying to fit a full tuning curve. However, it seems to me that his selection method excludes neurons tuned to straight-ahead as their slope should be flat for a symmetric range of headings.

Regression and correlational analyses are extremely sensitive to outliers. To be sure that the results are robust, please repeat the analyses after outlier correction, e.g., based on the 1.5xIQR rule along each dimension or based on each data points influence on the regression (Cook's distance).

Looking at the currently depicted sigmoids, I suggest increasing the maximal value for the lapse rate. Even though it is not fully clear to me what a lapse rate of neurometric curve actually means, achieving a better fit for the curves seems essential given that the PSEs are at the center of the results.

The methods section provides information about the number of neurons per area included in the analyses in general. Additional information about the number of neurons per neurometric curve would be useful.

Please add in l. 698 how the baseline activity for each neuron was determined, e.g., based on a single interval at the beginning of the session or based on recordings in between trials.

Probably just a typo but in l. 689 Pearson's correlation has nothing to do with linear regression.

Writing

Generally, I find the manuscript to be well-written and organized. However, in its current form, the manuscript is geared towards a small audience, electrophysiologists familiar with most publications by the Angelaki and DeAngelis labs. A much wider audience could be reached by 1) phrasing more precisely and 2) providing more information, reasoning, and explanations.

The verb 'recalibrate' is often used in a way that doesn't match the literature and the way recalibration is thought of. Information cannot be recalibrated; cues cannot recalibrate actively; responses do not recalibrate (together). A system is recalibrated and then it interprets incoming information differently. In most instances, a simple replacement of recalibrated with "shifted" will help -- AND/OR include a statement early on defining these more colloquial uses of recalibrate (e.g., "we refer to this pattern of neural activity as 'recalibration'…)

A few examples for more common and precise phrasing:

l. 35 -> recalibrating itself based on information from …

l. 36 -> estimates for subsequently presented unisensory stimuli are shifted towards each other.

l. 25 -> the tuning of neural responses to … cues was shifted in the same direction as the monkeys' perceptual judgments of subsequently presented unisensory stimuli.

l. 43 -> tuning of vestibular neurons was shifted in the same direction as vestibular heading perception.

This issue is present throughout the manuscript but especially so in the abstract, in brief section, and the discussion. I doubt that someone not familiar with the literature can understand the abstract. There is a section in the discussion (l. 483f) that is written in a very abstract manner but phrased according to the literature, i.e., in accordance with the ways most people think about recalibration. Please adjust the rest of the text accordingly.

Comments in chronological order:

l. 57 Behavioral papers cited in a sentence about the neuronal basis of multisensory integration.

l. 76 Just saying that results exist is rather unusual. In a few words, what did the neuroimaging studies find?

l. 91 Burge et al. is not about heading perception but about visual-haptic slant perception.

l. 103 Again, the typical phrasing would be that perception was recalibrated as indicated by shifts in subsequent perceptual estimates.

l. 108f It would be easier for readers to learn about supervised recalibration in the discussion, as now they have to shift mentally from unsupervised to supervised and back to unsupervised recalibration.

l. 119 Why would diverging shifts in the tuning of neural mechanisms between cortical areas not be detectable when both perceptual estimates are shifted in the same direction?

l. 126 Isn't that what the neuroimaging studies showed? Changes in relatively early areas?

l. 140 "therefore we expected to see perceptual shifts resulting from unsupervised recalibration in MSTd and PIVC". That is a surprising claim, which needs further explanation as it implies that perceptual decisions are based on neural activity in these relatively early areas. Maybe leave that conclusion entirely to the discussion.

l. 145 The claim that VIP underlies cognitive processing will startle many. Why not simply say that VIP seems to be involved in perceptual decision making or higher order perceptual functions some of which yet have to be understood?

l. 161 Please explain that 1) during the recalibration block, the monkeys are exposed to visual-vestibular stimulus pairs with a consistent discrepancy in heading direction and that 2) in pre- and post-recalibration blocks the perception of heading direction indicated by unisensory stimuli is measured and that 3) recalibration effects are measured as the difference between pre- and post-recalibration results. The text should make sense to a reader who has never performed a recalibration study. No reader should be forced to read another paper just to understand the most crucial aspects of the current one!

l. 181f It does not seem necessary to describe the figure in detail in the text. Explanations of how to read a figure are best placed in its caption. Given that these are selected-curves of a single-subject in one of several sessions, the size of the shifts should not be compared or discussed in the text.

l. 200 At this point the reader has never heard that the monkeys underwent several sessions nor in which way the sessions differed from each other. Please add that information.

l. 202 Why force the reader to learn what δ + means in this manuscript? It is much better to just speak of 'sessions in which visual heading was rightwards of vestibular heading during the recalibration phase'. Please apply that thought throughout.

l. 232 What does 'tuning recalibrate with perceptual shifts' mean? A good advice I got from a book on scientific writing is to take sentences literally even when they describe scientific matter. One possible general phrasing would be that the neural tuning shifted in the same (opposite for VIP) direction as heading perception. A more results-oriented phrasing is that the neurometric functions shifted in the same direction as the psychometric functions for each modality. Please repair this throughout the text!

l. 240 This description of the neurometric analysis is not sufficient (see comment in the methods section). In addition, "PSEs were extracted similar to" is confusing, PSEs always correspond to the 50% point of a psychometric function (e.g., the mean of the Gaussian distribution). More importantly, it should be explained to readers that the PSE of a neurometric curve is the physical heading direction at which the chances to make a rightward judgment based on the firing rates of the neurons are fifty-fifty, i.e., straight-ahead according to the neuronal data.

l. 245 I think this is the first instance in which the term "behavioral shifts" is used instead of "perceptual shifts" but then it occurs consistently and is even present in the figures. In psychophysics, the term "behavioral shifts" would be used if there is any reason to suspect that the behavior does not correspond to the percepts, e.g., because participants show a response bias rather than a perceptual bias. If this might be the case, it should be discussed, otherwise please use perceptual.

l. 286 Again, please phrase more precisely.

l. 315f I like the analysis and the paragraph could function as a prototype for the subsequent paragraphs regarding its degree of abstraction and briefness. Yet, again more precise phrasing would be nice, e.g., neurons don't respond visually they exclusively respond when visual stimuli are presented.

l. 334f Again, in my view it is not necessary to describe figures and do so panel by panel.

l. 339f The last sentence of the paragraph does not parse.

The claim that neural recalibration follows the velocity profile of the stimulus is too strong and not correct as the figures show correlations not neural recalibration. For the earlier areas, the claim that the significant correlation between neurometric and psychometric shifts is driven by firing rates during the period of maximal stimulus velocity might be correct.

l. 350f Similarly, for VIP neurons it is also not necessary to describe the figure.

l. 356 The claim that the correlations between neurometric and psychometric shifts given firing rates recorded at the end of the stimulus presentation has nothing to do with choice behavior but reflects neuronal recalibration is not substantiated at this point, an anti-correlation is a strong relation. The conclusion might be drawn based on the last section of the results.

l. 364 "During recalibration" means during the recalibration phase but I think that is not what the authors mean because there are no behavioral choices during the recalibration phase. Please search the text for this phrasing, it probably is not adequate at other instances, too.

l. 412 Not sure if this is the first instance, but cross-sensory is not a word used in the literature, please use either multisensory or cross-modal or across the senses. Some authors will point out that multisensory should only be used in the case of perceptual fusion. Please replace the word throughout the text.

l. 412f This holds for the full Discussion: See above comments, the phrasing is very uncommon, especially the use of the verb 'recalibrate'. Almost all instances of 'recalibrated' should be replaced with 'changed' or 'shifted'. "Together with" means "shifts in same direction as perceptual shifts" and so on. It gets much better from line 478 on.

l. 422 Not sure if vision scientists would call MSTd a multisensory area.

l. 432f Why does 'unsupervised' imply changes in early areas? I wondered the same in the introduction. And how does that go together with the suggestion that the conflict is detected in ACC? It might be easiest to just refer to the neuroimaging studies or simply not make a claim.

l. 457f What is "individualized recalibration"? The literature uses "modality-specific". I fail to see what this section adds that is not in the previous section. Why would the yoking found in the supervised recalibration study for both perceptual and neuronal shifts predict uniformity in the neuronal shifts in a paradigm that leads to non-yoked perceptual shifts?

l. 499 Please phrase the conclusion as a possibility, as it remains unclear to which degree supervised and unsupervised recalibration correspond.

l. 513 'Shift in reference frame' does this refer to a change in the supramodal definition of straight-ahead based on the vestibular tuning in lower sensory areas? The idea that VIP is tuned in a vestibular reference frame fits with earlier studies investigating visual-tactile reference frames (e.g., Avillac et al., Sereno and Huang, and also Graziano recording in F4).

l. 531 what does "cross-sensory recalibration, vs. simply reflecting the recalibrated signals" mean? What are recalibrated signals and why are they different from cross-modal recalibration?

General writing guidelines:

All figures should be optimized for the outlet, which in the case of *eLife* is wide and short figures as the text is never set in two-columns.

All acronyms should be defined when they are used for the first time, e.g., point of subjective equivalence (PSE). It can be very helpful for readers to treat abstract, significance statement, main text, and methods as separate and define acronyms anew.

A number and its unit are separated by a space, e.g., 300 ms instead of 300ms. Please check this throughout the text including the figures.

Figure 1

A: A visualization of the optic flow stimuli (e.g., two subsequent frames or little arrows indicating the motion vectors) would be nice.

B: It should be indicated either in the figure or in the caption that δ was constant within a single session, but theta was varied within a session and could take on all values depicted in A.

The grey vector corresponds to the combined direction if both cues have the same reliability, which probably wasn't exactly the case given Figure 2A.

C: 'no choice' is misplaced.

All of them, please add a space between a number and its unit.

Figure 2

A,B please indicate the monkey and the session number.

A,B usually rightward choice ratio would be interpreted as the ratio of rightward to leftward choices, I assume the proportion of rightward responses is shown.

C indicate how the shift was calculated.

C I cannot see the error bars referred to in the caption.

C please indicate the distribution of each monkey to assure readers that the results hold within and across subjects (see comments on the statistical analysis).

*Reviewer #2 (Recommendations for the authors):*

Aside from the major comments outlined in the public review, I found that some figure legends should be expanded. For instance, in Figure 8 it is unclear what the empty and filled circles indicate, respectively. I would recommend the authors to check the manuscript carefully.

While in general the manuscript is well written, I found the "in brief" section rather difficult and not suitable for a broader audience. I would suggest rewriting the section accordingly.

---

## [Author Response]

Essential revisions:1) Please report summary statistics (means and correlation coefficients) and p values for individual animals, for the data in Figure 2C, 3B, 4B, and 5B. This is not to say that the results must be significant at the individual animal level in order to support the study's main conclusions; we are all aware of the practicalities and accepted conventions of the field. Yet we feel it is important to be up front about this limitation, if indeed some of the individual monkey results fail to reach significance.To be clear, the reasoning behind this request is that repeated observations from the same subject/condition are not independent of each other, thus correlations calculated from pooled data might obscure, inflate, or even reverse the true relationship between the variables (e.g., Simpson's Paradox).Apart from individual subject analyses, another approach (indeed our strong recommendation) is to perform a hierarchical analysis. There are multiple options for this; probably the easiest one is to calculate linear mixed regression models (LMM) with one variable as predictor, the other as outcome and no intercept. This can be done in the frequentist way using (for example) the lmer package in R, or in a Bayesian way using Stan, which even as an automatized module for GLMMs. Partial correlations can be achieved by adding the variable that is partialized out as a predictor.In summary, the authors may choose one (or more) of the three mentioned approaches to enhance statistical rigor: (1) separate t-tests and correlations for each animal and condition, (2) frequentist hierarchical linear models, and (3) Bayesian hierarchical linear models. It is worth pointing out that Bayesian approaches have advantages over frequentist methods, for instance quantifying the evidence for the null hypothesis (and thus better evaluating negative results such as Figure 4B-right).

Thank you for raising this important point. In response, we 1) added summary statistics for the individual monkeys, and 2) applied the linear mixed model (LMM) analysis to the data, with neuronal shifts as the dependent variable, perceptual shifts as the fixed predictor, and monkeys as a random effects grouping factor. Results from both of these approaches indicate that the findings are consistent across monkeys. We added these results as supplementary tables (see source data associated with each figure) to the revised manuscript. The LMM analyses (with monkeys as a random effects factor) did not provide better fits than the pooled model (PM) analyses (with pooled data across monkeys). These additional analyses support the main findings of the manuscript.

2) Please consider applying a goodness-of-fit criterion to the neurometric curves before inclusion of their PSEs in the neuron-behavior correlation analyses – AND/OR evaluate the reliability of the PSEs using standard error obtained from the fitting procedure, or a bootstrap-based confidence interval. We would not require individual neuron shifts (δ-PSE) to be significant according to such SEs (i.e., through error propagation), but a reanalysis after removing particularly poor fits seems appropriate. The criterion to use for this is difficult for us to specify and thus can use your best judgment.

Thank you for this valid suggestion. To quantify the reliability of each neurometric PSE we used the standard deviation (SD) of the bootstrapped PSEs obtained from the fitting procedure. We considered that this measure would be a more direct estimate of PSE reliability (the parameter of interest) vs. more general goodness-of-fit measures (such as pseudo-R^2^) which are influenced by other parameters (e.g., thresholds and lapse rates) that are less relevant for assessing PSE estimate reliability. We required that the SD of the neuron’s bootstrapped PSE was < 10°, in both the pre-and post-recalibration blocks. This cutoff was chosen empirically, based on the distribution of SD values across neurons (see Figure 3—figure supplement 1). This screening was applied in addition to (after) the initial screening for significant tuning (as described in the original manuscript). We have now updated the manuscript to say that the neurons were first screened for significant response tuning, and then the remaining neurons were screened for neurometric PSE reliability (please see the revised Methods subsection “Neuronal shifts”, paragraph 1).

For MSTd, the neurometric PSE reliability screening removed 9 neurons from the vestibular data and 9 neurons from the visual data. For PIVC, this removed 14 neurons from the vestibular data and 11 neurons from the visual data. For VIP, this removed 16 neurons from the vestibular data and 24 neurons from the visual data. We redid all the analyses including only the neurons that passed this additional screening (the results remained similar), updated the figures, and revised the manuscript accordingly.

3) Depending on how the first two points are handled, and the outcome thereof, it may be necessary to scrutinize the role of outliers in generating the correlation coefficients and p values obtained. For instance, is the correlation of Figure 3B-left still significant without the upper four points, and 3B-right without the rightmost three points? A hierarchical analysis and/or neurometric goodness-of-fit criterion could reduce the role of outliers, in which case no formal outlier correction or other way to address this is needed.

The robustness of the results was assessed with the systematic approaches described above (in response to the two previous comments). Namely: additional screening to remove neurons with low-reliability estimates for the neuronal PSE (see the response to Comment #2), and additional analysis (LMM) to account for the random effects of different monkeys (see the response to Comment #1). These specific data points passed this screening, and the results were robust to further analyses that took into account the random effects of monkeys. In addition, we tested the influence of outliers using ‘Cook's distance’ (Author response image 1) . Redoing the analyses after removing data points with Cook’s distances larger than three times the mean provided similar results as before (summarized in Author response table 1).

**Author response image 1. sa2fig1:** Cook's distances. Cook’s distances for the vestibular and visual data (top and bottom row, respectively) from areas (A) MSTd, (B) PIVC, and (C) VIP. The green dashed line marks three times the mean Cook's distance.

**Author response table 1. sa2table1:** ‘Cook's distance’ outlier analysis.

MSTd	r	p	N	
Vestibular	All	0.62	0.019 *	14
	Outliers Removed	0.63	0.021 *	13
Visual	All	0.38	2.7 × 10^-3^ ***	59
	Outliers Removed	0.36	7.8 × 10^-3^ ***	53
PIVC	r	p	N	
Vestibular	All	0.80	9.7 × 10^-8^ ***	30
	Outliers Removed	0.81	9.3 × 10^-8^ ***	29
Visual	All	0.26	0.47	10
	Outliers Removed	0.26	0.47	10
VIP	r	p	N	
Vestibular	All	0.77	2.7× 10^-8^ ***	37
	Outliers Removed	0.80	1.1× 10^-8^ ***	35
Visual	All	-0.68	8.4× 10^-7^ ***	42
	Outliers Removed	-0.70	8.0 × 10^-7^ ***	38

N = number of neurons, r, and p-values from Pearson correlations. *** p < 0.001; * p < 0.05.

Reviewer #1 (Recommendations for the authors):The results are really interesting, yet, the manuscript in its current form needs revisions along two dimensions, 1) data analysis and 2) writing.MethodsI am aware that basically all analyses in this manuscript have been used in published papers. The problems outlined below hold for those papers, too, sorry. Bad (or good) luck having this time a reviewer from another field.Correlations derived from data that includes multiple repetitions per subject (and condition) require a hierarchical analysis because repeated observations from the same observer are not independent of each other. Subject A might on average have higher shifts (behavioral and neurometric) than subject B. In that case, a non-hierarchical analysis might return a significant positive correlation even if within subject A negative behavioral shifts are associated with positive neurometric shifts and vice versa. There are multiple ways to account for intra-subject correlations, probably the easiest one is to calculate linear mixed regression models (LMM) with one variable as predictor, the other as outcome and no intercept. This can be done in the frequentist way using the lmer package in R and in a Bayesian way using Stan which even as an automatized module for GLMMs. Partial correlations can be achieved by adding the variable that is partialized out as a predictor. Of course, it is also possible to just analyze each subject separately.

Thank you for raising this valid point. In response, we: (1) calculated the individual monkey summary statistics, and report these in the revised manuscript (Figure 2–source data 1 presents the behavioral shift data; Figure 3–source data 1, Figure 4–source data 1, and Figure 5–source data 1 present the correlations between neuronal and behavioral shifts for MSTd, PIVC, and VIP, respectively). (2) We performed the linear mixed model (LMM) analysis, and present the results, and comparison thereof to the pooled model (PM) with pooled data across monkeys, in the respective supplementary tables (see source data). Please see our response to Essential Revision #1 (above) for further details.

The same holds for any comparisons of mean values (across heading offsets during the recalibration phase or across modalities). For example, a t-test that includes perceptual shifts from all sessions and all monkeys is not valid as the values from a single monkey are correlated. Again, there are several ways to account for this. For example, an LMM with cross-modal discrepancy as predictor (and a free intercept) or each monkey's data could be tested separately.The procedure to derive neurometric curves, which are central to the study, should be explained better. In the main text and in the figures, it should be clarified that each data point shows the proportion of trials in which an ideal observer would make a rightwards choice given only the firing rates of the neurons (and assumed anti-neurons). In the methods section, it should be explained in detail how the ROC curve is derived. Readers unfamiliar with the method should understand that the ROC incorporates decisions for a range of decision thresholds and that the area under the ROC curve (AUROC, not ROC value) corresponds to a theoretical observer's ability to discriminate between leftwards and rightwards motion given firing rates across repetitions of a specific motion direction. Finally, those AUROC values (ranging from 0.5 to 1) are mapped onto the probability of a rightwards choice (ranging from 0 to 1) given a heading direction.

In response to this comment, we added more detailed explanations about the procedure to derive the neurometric curves (please see the revised Methods subsection “Neurometrics”, paragraphs 1-2). We also added to the figure legends (regarding neurometric curves) that each data point shows the proportion of trials in which an ideal observer would make a rightward choice given the firing rates of the neurons. We would like to clarify that for this calculation we did not compare neurons to “anti-neurons” (which would assume PSE = 0°). Rather, we z-scored the data in reference to the *pre-calibration* firing rates. Using a common reference (pre-recalibration mean, corresponding to z-score = 0) to normalize both pre- and post-recalibration data was needed to allow for and expose PSE shifts. Then we calculated ROC curves by moving a ‘criterion’ value from the minimum to the maximum z-score (in 100 steps), and plotting the probability that the z-scores exceeded the criterion vs. whether z-score = 0 (the pre-recalibration mean) exceeded that same criterion, or not (1 or 0, respectively). We have also added this clarification to the revised manuscript (see Methods subsection “Neurometrics”, paragraph 2).

The use of PSEs for the correlational analysis should be conditional on the goodness of fit of the sigmoid and the SD should fall within a reasonable range. A PSE derived from an ill-fitting or very flat curve has no informational value and is likely to be extreme, which in turn is huge a problem for any correlational analysis.

In response to this comment we further screened the neurons by calculating the SD of the neurometric PSE (from bootstrapped values) and excluded neurons with SDs > 10° (either pre- or post-recalibration). We considered that this would be a more direct measure of PSE reliability (the parameter of interest) vs. goodness-of-fit of the whole psychometric function, which could be affected by other parameters (such as thresholds and lapse rates). Please see our response to Essential Revision #2 (above) for further details.

The methods section states that only neurons tuned to heading directions as indicated by cues in a specific modality were included in the neurometric analysis for that modality. If that exclusion criterion was applied, why are the neurometric curves for 'the other modality' so bad? I would have expected that neurons tuned to visual heading cues result in a good neurometric function for visual heading no matter where these neurons are located in the brain. In turn, I would have expected flat neurometric functions for visual heading direction in PIVC if also neurons tuned to vestibular heading information were included in the analysis. What am I missing?

In the revised Methods subsection “Neuronal shifts”, paragraph 1, we now better explain the inclusion criteria for the different stages of analysis. First, we calculated neurometric functions for all recorded cells, and both modalities, whether or not the cell was significantly tuned. For the subsequent neuronal vs. behavioral shift analyses, we only included neurons that satisfied both the following criteria (the second criterion was added to the revised manuscript in response to Essential Revisions, Comment #2): i) significant tuning to the corresponding cue, and ii) reliable neurometric PSE estimates (SD < 10°, both in the pre- and post-recalibration blocks). The example neuron from PIVC in Figure 4 responded significantly to vestibular (but not visual) stimuli. Thus, although neurometric fits are presented for both cues in Figure 4, this PIVC neuron was included only in the vestibular (but not the visual) PSE correlation analysis.

Another instance of me probably not getting something: to identify neurons tuned for heading directions as cued by one of the two modalities, spike rates were regressed onto heading directions and only those neurons with a significant slope were included. I get that the range of tested headings is smaller than the width of a typical tuning curve and thus a linear regression makes more sense than trying to fit a full tuning curve. However, it seems to me that his selection method excludes neurons tuned to straight-ahead as their slope should be flat for a symmetric range of headings.

Indeed our selection criteria were selective for neurons that have sloped tuning around straight-ahead, and exclude neurons that could be tuned to forward motion stimuli (i.e., with a tuning preference straight-ahead). The reasons for this are: (1) neurons with sloped tuning straight-ahead are most informative for heading discrimination (large Fisher information, Gu et al., 2010). By contrast, neurons with a tuning preference to straight-ahead stimuli have relatively flat responses around straight-ahead (low Fisher information) and are thus less informative for heading discrimination. Accordingly, small shifts can be readily detected in neurons with sloped tuning (but not in those with flat tuning) around straight ahead. (2) In the cortical areas of interest in this study, a disproportionately large number of neurons have steep tuning slopes around straight ahead, presumably to support heading discrimination (Chen et al., 2011b; Gu et al., 2010). (3) The standard neurometric analysis wouldn’t work for neurons with a tuning preference to straight-ahead (rightward and leftward headings around straight-ahead would elicit ambiguous firing rates). Thus estimating tuning shifts in these neurons would require different (heterogeneous and more complex) models, and the results would be very noisy. Therefore, in this study, we focused on the prevalent neurons with sloped tuning around straight-ahead. We have added these points to the revised Methods subsection “Data analysis”, paragraph 3.

Regression and correlational analyses are extremely sensitive to outliers. To be sure that the results are robust, please repeat the analyses after outlier correction, e.g., based on the 1.5xIQR rule along each dimension or based on each data points influence on the regression (Cook's distance).

In this revision, we added: 1) summary statistics of the individual monkeys, and performed an LMM analysis (in response to Essential Revisions, Comment #1), and 2) we screened the neurons based on their PSE estimate reliability (in response to Essential Revisions, Comment #2). In addition, we also tested the influence of outliers using ‘Cook's distance’ (see details in response to Essential Revisions, Comment #3). Results from these analyses support the robustness of the results.

Looking at the currently depicted sigmoids, I suggest increasing the maximal value for the lapse rate. Even though it is not fully clear to me what a lapse rate of neurometric curve actually means, achieving a better fit for the curves seems essential given that the PSEs are at the center of the results.

We assessed the effect of allowing for a broader range of lapse rates on the neurometric fits, and PSEs in particular. Allowing for a broader range of lapse rates increased pseudo-R^2^ values (expected when increasing the number of model parameters) and sometimes affected threshold estimates (thresholds could be under- or overestimated, depending on whether lapse rates are allowed or not, respectively). However, it had little influence on the PSE estimates. For example, the post-recalibration neurometric curve in Figure 3A (blue; redrawn in Author response image 2) has a threshold = 17.8°. If larger lapse rates are allowed (Author response image 2), this could provide an unreasonably low threshold estimate (threshold = 0.6°). Thus, allowing (or not) large lapse rates can drastically affect threshold estimates. By contrast, the post-recalibration neurometric PSE was 3.6° and 2.8° for the two curves, respectively. Hence, in general, the effect of lapse rates on PSE estimates is small. Because PSE (not threshold) is the parameter of interest in this study, we opted for a more conservative neuromeric fit (with fewer parameters, i.e. tighter lapse rates) and exclusively used the PSE (not threshold) estimates. In addition, we added screening of neuronal PSE estimate reliability (via bootstrapping) in the revised manuscript (please see details in response to Essential Revisions, Comment #2).

**Author response image 2. sa2fig2:** Influence of lapse rates on example neurometric fit. (A) Neurometric fit of vestibular responses, without allowing large lapse rates, for the example MSTd neuron (same as Figure 3A). (B) Neurometric fit of the same data, allowing large lapse rates.

The methods section provides information about the number of neurons per area included in the analyses in general. Additional information about the number of neurons per neurometric curve would be useful.

We added this information in the Methods subsection “Neuronal shifts”, paragraph 1.

Please add in l. 698 how the baseline activity for each neuron was determined, e.g., based on a single interval at the beginning of the session or based on recordings in between trials.

Baseline FRs were calculated by taking the average FR in the 1 s window before stimulus onset. We have added this to the revised Methods, subsection “Data analysis”, paragraph 2.

Probably just a typo but in l. 689 Pearson's correlation has nothing to do with linear regression.

We have corrected this.

WritingGenerally, I find the manuscript to be well-written and organized. However, in its current form, the manuscript is geared towards a small audience, electrophysiologists familiar with most publications by the Angelaki and DeAngelis labs. A much wider audience could be reached by 1) phrasing more precisely and 2) providing more information, reasoning, and explanations.

In response to this comment and thanks to the comments below, we now provide more information and explanations, with improved phrasing in our revised manuscript.

The verb 'recalibrate' is often used in a way that doesn't match the literature and the way recalibration is thought of. Information cannot be recalibrated; cues cannot recalibrate actively; responses do not recalibrate (together). A system is recalibrated and then it interprets incoming information differently. In most instances, a simple replacement of recalibrated with "shifted" will help -- AND/OR include a statement early on defining these more colloquial uses of recalibrate (e.g., "we refer to this pattern of neural activity as 'recalibration'…)

In the revised manuscript we now use the term ‘recalibrate’ more carefully. This entailed replacing many instances of “recalibrated” with “shifted”.

A few examples for more common and precise phrasing:l. 35 -> recalibrating itself based on information from …

Thank you for the improved phrasing. We have updated the text.

l. 36 -> estimates for subsequently presented unisensory stimuli are shifted towards each other.

Thank you for the improved phrasing. We have updated that sentence.

l. 25 -> the tuning of neural responses to … cues was shifted in the same direction as the monkeys' perceptual judgments of subsequently presented unisensory stimuli.

Thank you for the improved phrasing. We have updated that sentence.

l. 43 -> tuning of vestibular neurons was shifted in the same direction as vestibular heading perception.This issue is present throughout the manuscript but especially so in the abstract, in brief section, and the discussion. I doubt that someone not familiar with the literature can understand the abstract. There is a section in the discussion (l. 483f) that is written in a very abstract manner but phrased according to the literature, i.e., in accordance with the ways most people think about recalibration. Please adjust the rest of the text accordingly.

Thanks for bringing this to our attention. We have updated the relevant text accordingly.

Comments in chronological order:l. 57 Behavioral papers cited in a sentence about the neuronal basis of multisensory integration.

We updated those references

l. 76 Just saying that results exist is rather unusual. In a few words, what did the neuroimaging studies find?

We agree and have now added a brief explanation of these neuroimaging studies in the revised Introduction.

l. 91 Burge et al. is not about heading perception but about visual-haptic slant perception.

Thank you. We removed this reference from the paper.

l. 103 Again, the typical phrasing would be that perception was recalibrated as indicated by shifts in subsequent perceptual estimates.

Thanks. We have updated that sentence.

l. 108f It would be easier for readers to learn about supervised recalibration in the discussion, as now they have to shift mentally from unsupervised to supervised and back to unsupervised recalibration.

We agree and have moved this paragraph about supervised recalibration to the Discussion.

l. 119 Why would diverging shifts in the tuning of neural mechanisms between cortical areas not be detectable when both perceptual estimates are shifted in the same direction?

We now better clarify this point in the revised manuscript. In the supervised calibration paradigm, the overall shifts for both vestibular and visual cues were “yoked” in the same direction. Thus, we could not dissociate whether neuronal shifts for a particular cue (e.g., vestibular) follow the behavioral shifts for that cue (vestibular) or the other cue (visual). Both predict shifts in the same direction, and it would be difficult to dissociate these based on small differences in the expected shift magnitudes (because of noise). This (unsupervised) paradigm elicits behavioral shifts in opposite directions, and thus can more readily discern if the vestibular neurometrics shift with visual (rather than vestibular) behavioral shifts. We have added this explanation to the revised Discussion subsection “Modality-specific recalibration of vestibular and visual cues”, paragraph 3.

l. 126 Isn't that what the neuroimaging studies showed? Changes in relatively early areas?

Yes, we have updated the sentence by referencing neuroimaging studies that found auditory-visual recalibration in relatively early sensory areas (Amedi et al., 2002; Zierul et al., 2017).

l. 140 "therefore we expected to see perceptual shifts resulting from unsupervised recalibration in MSTd and PIVC". That is a surprising claim, which needs further explanation as it implies that perceptual decisions are based on neural activity in these relatively early areas. Maybe leave that conclusion entirely to the discussion.

We now justify this hypothesis based on the neuroimaging studies that found auditory-visual recalibration in relatively early sensory areas. Also, we further explain that because unsupervised plasticity is sensory-driven (occurs as a result of the cross-modal discrepancy, in the absence of overt feedback) we expected to observe neural correlates of recalibration in these lower-level sensory areas, and for these to propagate along the perceptual decision-making hierarchy.

l. 145 The claim that VIP underlies cognitive processing will startle many. Why not simply say that VIP seems to be involved in perceptual decision making or higher order perceptual functions some of which yet have to be understood?

We thank the reviewer for this refinement, and have updated that explanation accordingly.

l. 161 Please explain that 1) during the recalibration block, the monkeys are exposed to visual-vestibular stimulus pairs with a consistent discrepancy in heading direction and that 2) in pre- and post-recalibration blocks the perception of heading direction indicated by unisensory stimuli is measured and that 3) recalibration effects are measured as the difference between pre- and post-recalibration results. The text should make sense to a reader who has never performed a recalibration study. No reader should be forced to read another paper just to understand the most crucial aspects of the current one!

We agree and have now added more details to explain the experimental procedure.

l. 181f It does not seem necessary to describe the figure in detail in the text. Explanations of how to read a figure are best placed in its caption. Given that these are selected-curves of a single-subject in one of several sessions, the size of the shifts should not be compared or discussed in the text.

We have removed the unnecessary details from the text, and also removed the shift size comparison for the example plots. We also modified the figure legends.

l. 200 At this point the reader has never heard that the monkeys underwent several sessions nor in which way the sessions differed from each other. Please add that information.

Thank you. We have now added information (specifically the number of sessions, and how the sessions differed) so that the reader can better understand the results which follow (please see updated Results subsection “Vestibular and visual perceptual estimates shift toward each other”, paragraph 3).

l. 202 Why force the reader to learn what δ + means in this manuscript? It is much better to just speak of 'sessions in which visual heading was rightwards of vestibular heading during the recalibration phase'. Please apply that thought throughout.

We have now added (in response to the Reviewer’s previous comment) a better explanation regarding the two types of sessions, and the meaning of δ + and δ -. Because these appear numerously throughout the text, and there is not enough space in the figures to write explicitly 'sessions in which visual heading was rightwards of vestibular heading during the recalibration phase’ etc., we think that it is clearer to keep this nomenclature.

l. 232 What does 'tuning recalibrate with perceptual shifts' mean? A good advice I got from a book on scientific writing is to take sentences literally even when they describe scientific matter. One possible general phrasing would be that the neural tuning shifted in the same (opposite for VIP) direction as heading perception. A more results-oriented phrasing is that the neurometric functions shifted in the same direction as the psychometric functions for each modality. Please repair this throughout the text!

We have amended the language accordingly.

l. 240 This description of the neurometric analysis is not sufficient (see comment in the methods section). In addition, "PSEs were extracted similar to" is confusing, PSEs always correspond to the 50% point of a psychometric function (e.g., the mean of the Gaussian distribution). More importantly, it should be explained to readers that the PSE of a neurometric curve is the physical heading direction at which the chances to make a rightward judgment based on the firing rates of the neurons are fifty-fifty, i.e., straight-ahead according to the neuronal data.

We have updated the text accordingly.

l. 245 I think this is the first instance in which the term "behavioral shifts" is used instead of "perceptual shifts" but then it occurs consistently and is even present in the figures. In psychophysics, the term "behavioral shifts" would be used if there is any reason to suspect that the behavior does not correspond to the percepts, e.g., because participants show a response bias rather than a perceptual bias. If this might be the case, it should be discussed, otherwise please use perceptual.

We agree with the reviewer’s suggestion. We have replaced the term "behavioral shifts" with "perceptual shifts".

l. 286 Again, please phrase more precisely.

We now changed this subtitle to “Neuronal tuning in VIP to both vestibular and visual stimuli shifted according to vestibular perceptual shifts”.

l. 315f I like the analysis and the paragraph could function as a prototype for the subsequent paragraphs regarding its degree of abstraction and briefness. Yet, again more precise phrasing would be nice, e.g., neurons don't respond visually they exclusively respond when visual stimuli are presented.

We have revised the phrasing accordingly.

l. 334f Again, in my view it is not necessary to describe figures and do so panel by panel.

We have removed unnecessary descriptions of the figure from the main text.

l. 339f The last sentence of the paragraph does not parse.The claim that neural recalibration follows the velocity profile of the stimulus is too strong and not correct as the figures show correlations not neural recalibration. For the earlier areas, the claim that the significant correlation between neurometric and psychometric shifts is driven by firing rates during the period of maximal stimulus velocity might be correct.

We agree with this point and have refined the explanation in line with the reviewer’s suggestion.

l. 350f Similarly, for VIP neurons it is also not necessary to describe the figure.

Thanks, we have refined this paragraph.

l. 356 The claim that the correlations between neurometric and psychometric shifts given firing rates recorded at the end of the stimulus presentation has nothing to do with choice behavior but reflects neuronal recalibration is not substantiated at this point, an anti-correlation is a strong relation. The conclusion might be drawn based on the last section of the results.

We have removed that sentence.

l. 364 "During recalibration" means during the recalibration phase but I think that is not what the authors mean because there are no behavioral choices during the recalibration phase. Please search the text for this phrasing, it probably is not adequate at other instances, too.

We agree with the reviewer’s suggestion. We have replaced the phrase "during recalibration" with "after recalibration" where appropriate in the text.

l. 412 Not sure if this is the first instance, but cross-sensory is not a word used in the literature, please use either multisensory or cross-modal or across the senses. Some authors will point out that multisensory should only be used in the case of perceptual fusion. Please replace the word throughout the text.

We agree with the reviewer’s suggestion. We have replaced the phrase "cross-sensory " with " cross-modal" throughout the text.

l. 412f This holds for the full Discussion: See above comments, the phrasing is very uncommon, especially the use of the verb 'recalibrate'. Almost all instances of 'recalibrated' should be replaced with 'changed' or 'shifted'. "Together with" means "shifts in same direction as perceptual shifts" and so on. It gets much better from line 478 on.

We agree with the reviewer’s suggestion. We replaced many instances of the word “recalibrate” with “shifted”. We also replaced phrases like “recalibrated together with the corresponding vestibular perceptual shifts” with “shifted in the same direction as vestibular perceptual shifts” etc.

l. 422 Not sure if vision scientists would call MSTd a multisensory area.

Although MSTd is part of the extra-striate visual cortex, it has been extensively shown to have multisensory (including vestibular) responses. We back up this claim in the manuscript with relevant references.

l. 432f Why does 'unsupervised' imply changes in early areas? I wondered the same in the introduction. And how does that go together with the suggestion that the conflict is detected in ACC? It might be easiest to just refer to the neuroimaging studies or simply not make a claim.

We now justify this hypothesis (in the revised Introduction) based on the neuroimaging studies, as recommended by the reviewer.

l. 457f What is "individualized recalibration"? The literature uses "modality-specific". I fail to see what this section adds that is not in the previous section. Why would the yoking found in the supervised recalibration study for both perceptual and neuronal shifts predict uniformity in the neuronal shifts in a paradigm that leads to non-yoked perceptual shifts?

We agree with the reviewer’s suggested phrasing and replaced "individualized recalibration" with "modality-specific recalibration". In the supervised calibration paradigm, the overall shifts for both vestibular and visual cues were “yoked” in the same direction. Thus, we could not dissociate whether neuronal shifts for a particular cue (e.g., vestibular) follow the behavioral shifts for that cue (vestibular) or the other cue (visual). Both predict shifts in the same direction, and it would be difficult to dissociate these based on the expected shift magnitudes (because of noise). This (unsupervised) paradigm elicits behavioral shifts in opposite directions, and thus can more readily discern if the vestibular neurometrics shift with visual (rather than vestibular) behavioral shifts.

l. 499 Please phrase the conclusion as a possibility, as it remains unclear to which degree supervised and unsupervised recalibration correspond.

We agree with the reviewer’s suggestion and have amended the language accordingly.

l. 513 'Shift in reference frame' does this refer to a change in the supramodal definition of straight-ahead based on the vestibular tuning in lower sensory areas? The idea that VIP is tuned in a vestibular reference frame fits with earlier studies investigating visual-tactile reference frames (e.g., Avillac et al., Sereno and Huang, and also Graziano recording in F4).

We do not know if this necessarily means “a change in the *supramodal* definition of straight-ahead based on the vestibular tuning in lower sensory areas”. We just point out that neuronal signals in VIP have been previously shown to be in head/body centered coordinates, and that the shifts we observe here are in line with that notion. We have clarified this section, and added these relevant references, in the revised Discussion subsection “Contrary recalibration in higher-level area VIP ” paragraph 3. Thank you.

l. 531 what does "cross-sensory recalibration, vs. simply reflecting the recalibrated signals" mean? What are recalibrated signals and why are they different from cross-modal recalibration?

By “cortical areas actively involved in cross-modal recalibration” we mean that they had an instrumental (causal) role in vestibular-visual recalibration. By “simply reflecting the recalibrated signals” we mean that they were not instrumental (causal) for recalibration, and thus only reflect signals that were recalibrated in/by other brain areas. We have now refined and better explained this idea in the revised Discussion.

General writing guidelines:All figures should be optimized for the outlet, which in the case of eLife is wide and short figures as the text is never set in two-columns.

We have changed the layout of Figures 3, 4, and 5 accordingly.

All acronyms should be defined when they are used for the first time, e.g., point of subjective equivalence (PSE). It can be very helpful for readers to treat abstract, significance statement, main text, and methods as separate and define acronyms anew.A number and its unit are separated by a space, e.g., 300 ms instead of 300ms. Please check this throughout the text including the figures.

We thank the reviewer for these suggestions, and have amended the text accordingly.

Figure 1A: A visualization of the optic flow stimuli (e.g., two subsequent frames or little arrows indicating the motion vectors) would be nice.

We added a schematic (rightmost in Figure 1A) to visualize the optic flow.

B: It should be indicated either in the figure or in the caption that δ was constant within a single session, but theta was varied within a session and could take on all values depicted in A.

We added these points to the legend of Figure 1.

The grey vector corresponds to the combined direction if both cues have the same reliability, which probably wasn't exactly the case given Figure 2A.

The grey vector was not meant to reflect the *perceived* ‘combined cue’, rather, it is just a convention for defining the stimuli (visual and vestibular stimuli were offset ± 5° relative to this). We have clarified this in the legend.

C: 'no choice' is misplaced.

Thanks. We have corrected this.

All of them, please add a space between a number and its unit.

OK. We have updated accordingly.

Figure 2A,B please indicate the monkey and the session number.

We added this information to the figure legend.

A,B usually rightward choice ratio would be interpreted as the ratio of rightward to leftward choices, I assume the proportion of rightward responses is shown.

Yes. We replaced "ratio" with "proportion" in the y-label.

C indicate how the shift was calculated.

We added to the legend how the shift was calculated.

C I cannot see the error bars referred to in the caption.

The error bars are indeed difficult to see because the SEs are small. To improve visibility, we reduced the triangle symbol size and made the error bars thicker.

C please indicate the distribution of each monkey to assure readers that the results hold within and across subjects (see comments on the statistical analysis).

We updated Figure 2C to display the distribution of each monkey (using different texture patterns). In addition, the summary statistics for each monkey are presented in Figure 2-source data 1.

Reviewer #2 (Recommendations for the authors):Aside from the major comments outlined in the public review, I found that some figure legends should be expanded. For instance, in Figure 8 it is unclear what the empty and filled circles indicate, respectively. I would recommend the authors to check the manuscript carefully.

We thank the reviewer for bringing this to our attention. Filled and empty circles indicate significant and non-significant partial correlations respectively. We have added this and other details to the figure legends.

While in general the manuscript is well written, I found the "in brief" section rather difficult and not suitable for a broader audience. I would suggest rewriting the section accordingly.

We removed the "in brief" and “highlights” sections from the manuscript because these are not standard in *eLife*.